# Principled Data Augmentation for Learning to Solve Quadratic Programming Problems

**Chendi Qian** **Christopher Morris**
Department of Computer Science
RWTH Aachen University
Aachen, Germany
`chendi.qian@log.rwth-aachen.de`

## Abstract

Linear and quadratic optimization are crucial in numerous real-world applications, ranging from training machine learning models to solving integer linear programs. Recently, learning-to-optimize methods (L2O) for linear (LPs) or quadratic programs (QPs) using message-passing graph neural networks (MPNNs) have gained traction, promising lightweight, data-driven proxies for solving such optimization problems. For example, they replace the costly computation of strong branching scores in branch-and-bound solvers, thereby reducing the need to solve many such optimization problems. However, robust L2O MPNNs remain challenging in data-scarce settings, especially when addressing complex optimization problems such as QPs. This work introduces a principled approach to data augmentation tailored for QPs via MPNNs. Our method leverages theoretically justified data augmentation techniques to generate diverse yet optimality-preserving instances. Furthermore, we integrate these augmentations into a self-supervised contrastive learning framework, thereby pretraining MPNNs for improved performance on L2O tasks. Extensive experiments demonstrate that our approach improves generalization in supervised scenarios and facilitates effective transfer learning to related optimization problems.

## 1 Introduction

Linear and quadratic optimization problems are fundamental problems in machine learning, operations research, and scientific computing [Boyd and Vandenberghe, 2004, Nocedal and Wright, 1999]. Many real-world applications, such as resource allocation, logistics, and training machine learning models, rely on efficiently solving large-scale *linear programming* (LPs) and *quadratic programming* (QPs). In addition, they play a key role in state-of-the-art integer-linear optimization solvers, allowing the computation of lower bounds, and are the basis for crucial heuristics such as *strong branching* for variable selection [Achterberg et al., 2005].

Recently, machine learning techniques, particularly *message-passing graph neural networks* (MPNNs) [Gilmer et al., 2017, Scarselli et al., 2008] have been explored for *learning-to-optimize* (L2O) approaches, aiming to learn to solve LPs and QPs in a data-driven fashion [Bengio et al., 2021, Cappart et al., 2023], enhancing solver efficiency and improving generalization across different problem instances. For example, Gasse et al. [2019] used MPNNs to imitate the costly strong branching score for variable selection in integer-linear optimization solvers, which requires solving many LPs during the solving process. However, training robust models for L2O remains challenging due to the scarcity of labeled data, especially for complex optimization formulations such as QPs.

In response to this challenge, *self-supervised learning* (SSL) has emerged as a powerful paradigm for pretraining models on large, unlabeled datasets [Liu et al., 2021]. *Contrastive learning*, a key approach in SSL, has demonstrated significant success in graph-based tasks by leveraging data augmentation

39th Conference on Neural Information Processing Systems (NeurIPS 2025).

techniques to create diverse training examples You et al. [2020]. Despite advances in contrastive learning and L2O with MPNNs, a gap exists in integrating these two paradigms for LPs and QPs. Additionally, designing effective and principled data augmentation strategies for LPs and QPs remains non-trivial due to the structural constraints and optimality conditions inherent in these problems.

**Present work** In this work, we propose a principled approach to data augmentation for LPs and QPs, specifically designed to enhance the performance of MPNN-based L2O methods for such problems; see Fig. 1 for a high-level overview. Concretely, our contributions are as follows:

1. We introduce a set of novel and theoretically grounded augmentation techniques for LPs and QPs that preserve optimality while generating diverse training instances;

2. We apply these augmentations in both supervised and self-supervised settings, including contrastive pretraining of MPNNs to enhance downstream performance;

3. We empirically evaluate our approach on synthetic and benchmark datasets, showing that pretraining with our augmentations improves generalization and transferability across problem classes.

*By bridging the gap between data augmentation and L2O for LPs and QPs, our work offers a new perspective on enhancing neural solvers through supervised learning and self-supervised pretraining. The proposed augmentations improve generalization in data-scarce settings, enabling more efficient and robust learning-based optimization methods.*

## 1.1 Related work

This section reviews relevant literature on L2O, graph data augmentation, graph contrastive learning, and synthetic instance generation for LP and MILP problems.

**MPNN and L2O** Message Passing Neural Networks (MPNNs) [Gilmer et al., 2017, Scarselli et al., 2008] have been extensively studied and are broadly categorized into spatial and spectral variants. Spatial MPNNs [Bresson and Laurent, 2017, Duvenaud et al., 2015, Hamilton et al., 2017, Veličković et al., 2017, Xu et al., 2018] follow the message-passing paradigm introduced by Gilmer et al. [2017]. MPNNs have shown strong potential in learning to optimize (L2O). A widely adopted approach represents MILPs using constraint-variable bipartite graphs [Chen et al., 2022, Ding et al., 2020, Gasse et al., 2019, Khalil et al., 2022, Qian et al., 2024a]. Recent work has also aligned MPNNs with various optimization algorithms, including interior-point methods (IPMs) [Qian et al., 2024a, Qian and Morris, 2025], primal-dual hybrid gradient (PDHG) [Li et al., 2024a], and distributed algorithms [Li et al., 2024b, 2025]. From a theoretical perspective, several studies have analyzed the expressiveness of MPNNs in approximating solutions to linear and quadratic programming [Chen et al., 2022, 2023, 2024b,a, Wu et al., 2024].

**Graph data augmentation** Graph data augmentation is central to learning-based optimization, especially under data scarcity. Common strategies include feature-wise perturbations [You et al., 2020, Zhu et al., 2020, Suresh et al., 2021, Zhu et al., 2021, Thakoor et al., 2021, Bielak et al., 2022, Hu et al., 2023], structure-wise modifications such as node dropping and edge perturbation [Rong et al., 2019, Hassani and Khasahmadi, 2020, Zhu et al., 2020, Qiu et al., 2020, Papp et al., 2021, Zhu et al., 2021, Bielak et al., 2022, Hu et al., 2023], and spectral-domain augmentations [Yang et al., 2024a, Liu et al., 2022a, Lin et al., 2022, Wan et al., 2024], the latter aligning with recent works like S3GCL. Learning-based approaches [Yang et al., 2020, Zhao et al., 2021, Liu et al., 2022b, Yin et al., 2022, You et al., 2021, Wu et al., 2023] further enable adaptive augmentation via trainable modules. Complementary perspectives include graph rewiring [Topping et al., 2021, Karhadkar et al., 2022, Arnaiz-Rodríguez et al., 2022, Qian et al., 2023, Gutteridge et al., 2023, Barbero et al., 2023, Qian et al., 2024b] and structure learning [Jin et al., 2020, Liu et al., 2022c, Zou et al., 2023, Zhou et al., 2023, Fatemi et al., 2023], both of which can be viewed as task-specific augmentation techniques. For a broader overview, including robustness and self-supervised settings, see [Zhao et al., 2022, Ding et al., 2022].

**Graph contrastive learning** Graph augmentations also play a central role in graph self-supervised learning (SSL), which aims to learn transferable representations without labeled supervision. Graph SSL methods are categorized into contrastive, generative, and predictive approaches [Wu et al., 2021].

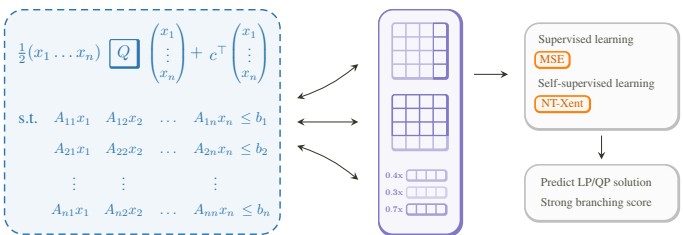

Figure 1: Overview of our framework for principled data augmentation for quadratic optimization problems. Given a QP instance, we apply transformations (e.g., adding/removing/scaling variables or constraints; see Section 2.2) to generate new instances, thereby augmenting the training dataset. We then use standard supervised learning or contrastive learning to train an MPNN.

We focus on contrastive ones, which operate at three levels: *global-to-global* (G2G), *local-to-local* (L2L), and *local-to-global* (L2G). G2G methods generate contrasting views of entire graphs, as in GraphCL [You et al., 2020] with structural perturbations, extended by CSSL [Zeng and Xie, 2021], GCC [Qiu et al., 2020] using random walks, AutoGCL [Yin et al., 2022] with learned augmentations, and AD-GCL [Suresh et al., 2021] with an adversarial objective. The only contrastive method targeting LPs is Li et al. [2024c], which adopts a CLIP-style [Radford et al., 2021] formulation. L2L methods contrast node-level views, e.g., GRACE [Zhu et al., 2020], GCA [Zhu et al., 2021], Graph Barlow Twins [Bielak et al., 2022], BGRL [Thakoor et al., 2021], and REGCL [Ji et al., 2024], with S3GCL [Wan et al., 2024] introducing spectral views. L2G methods such as DGI [Veličković et al., 2018], MVGRL [Hassani and Khasahmadi, 2020], and InfoGraph [Sun et al., 2019] contrast local and global representations to maximize mutual information.

While general-purpose augmentations are well-studied, few works target L2O problems. Duan et al. [2022] designs satisfiability-preserving augmentations for SAT problems, and Huang et al. [2023] samples neighborhoods around expert solutions. A concurrent work [Zeng et al., 2025] explores constraint permutations in MILPs. Compared with these works, we propose efficient, principled transformations, tailored for supervised and contrastive learning in linear and quadratic programming.

**Instance generation**  While random instance generators exist for LPs and MILPs [Gasse et al., 2019, 2022], data scarcity has driven model-based approaches, including stress testing [Bowly, 2019], VAEs [Geng et al., 2023], and diffusion models [Zhang et al., 2024]. Other works reconstruct or adapt instances [Wang et al., 2023, Guo et al., 2024, Yang et al., 2024c], or generate them from code or substructures [Li et al., 2024c, Liu et al., 2024]. In contrast, our approach is model-free, mathematically principled, and generates new instances through transformations with analytically tractable solutions.

## 1.2 Background

We introduce notations and define MPNNs, LPs, and QPs in the following.

**Notations**  Let $\mathbb{N} := \{0, 1, 2, \dots\}$. For $n \geq 1$, let $[n] := \{1, \dots, n\} \subset \mathbb{N}$. We use $\{\!\{\dots\}\!\}$ to denote multisets, i.e., the generalization of sets allowing for multiple instances of each of its elements. A *graph* $G$ is a pair $(V(G), E(G))$ with *finite* sets of *vertices* or *nodes* $V(G)$ and *edges* $E(G) \subseteq \{\!\{u, v\} \subseteq V(G) \times V(G) \mid u \neq v\}$. For ease of notation, we denote the edge $\{u, v\}$ in $E(G)$ by $(u, v)$ or $(v, u)$. The *neighborhood* of a node $v$ is denoted by $N(v) = \{u \in V \mid (v, u) \in E\}$, and its *degree* is $|N(v)|$. An *attributed graph* augments each node $v \in V$ with a feature vector $\sigma(v) \in \mathbb{R}^d$, yielding a node feature matrix $\boldsymbol{H} \in \mathbb{R}^{n \times d}$ where $\boldsymbol{H}_v = \sigma(v)$. The adjacency matrix of $G$ is denoted $\boldsymbol{A} \in \{0, 1\}^{n \times n}$, with $A_{ij} = 1$ if and only if $(i, j) \in E$. Vectors $\boldsymbol{x} \in \mathbb{R}^d$ are column vectors by default.

**LCQP**  In this work, we focus on special QPs, namely *linearly-constrained quadratic programming* (LCQP), of the following form,

$$\min_{\boldsymbol{x} \in \mathbb{R}^n} \tfrac{1}{2} \boldsymbol{x}^\intercal \boldsymbol{Q} \boldsymbol{x} + \boldsymbol{c}^\intercal \boldsymbol{x} \text{ s.t. } \boldsymbol{A} \boldsymbol{x} \leq \boldsymbol{b}. \tag{1}$$

Here, an LCQP instance $I$ is a tuple $(\boldsymbol{Q}, \boldsymbol{A}, \boldsymbol{b}, \boldsymbol{c})$, where $\boldsymbol{Q} \in \mathbb{R}^{n \times n}$ and $\boldsymbol{c} \in \mathbb{R}^n$ are the quadratic and linear coefficients of the objective, $\boldsymbol{A} \in \mathbb{R}^{m \times n}$, and $\boldsymbol{b} \in \mathbb{R}^m$ form the inequality constraints.

Note that the bounds of variables, e.g., $\boldsymbol{x} \geq \boldsymbol{0}$, can also be merged into the constraints. We assume that the smmetric and quadratic matrix $\boldsymbol{Q}$ is *positive definite* (PD), denoted as $\boldsymbol{Q} \succ \boldsymbol{0}$, so that the problem is convex and has a unique solution. The optimal solution $\boldsymbol{x}^*$ is a feasible solution such that $\frac{1}{2}\boldsymbol{x}^\mathsf{T}\boldsymbol{Q}\boldsymbol{x} + \boldsymbol{c}^\mathsf{T}\boldsymbol{x} \geq \frac{1}{2}\boldsymbol{x}^{*\mathsf{T}}\boldsymbol{Q}\boldsymbol{x}^* + \boldsymbol{c}^\mathsf{T}\boldsymbol{x}^*$, for any feasible $\boldsymbol{x}$. The corresponding dual variables for the constraints are denoted as $\boldsymbol{\lambda}^*$.[1] By setting the matrix $\boldsymbol{Q}$ to a zero matrix, we arrive at LPs,

$$\min_{\boldsymbol{x}\in\mathbb{R}^n} \boldsymbol{c}^\mathsf{T}\boldsymbol{x} \text{ s.t. } \boldsymbol{A}\boldsymbol{x} \leq \boldsymbol{b}. \tag{2}$$

Considering the QP of Eq. (1), the optimal primal-dual solutions must satisfy the *Karush–Kuhn–Tucker (KKT)* conditions [Nocedal and Wright, 1999, p. 321],

$$\boldsymbol{Q}\boldsymbol{x}^* + \boldsymbol{A}^\mathsf{T}\boldsymbol{\lambda}^* + \boldsymbol{c} = \boldsymbol{0} \tag{3a}$$

$$\boldsymbol{A}\boldsymbol{x}^* \leq \boldsymbol{b} \tag{3b}$$

$$\boldsymbol{\lambda}^* \geq \boldsymbol{0} \tag{3c}$$

$$\lambda_i^* \left(\boldsymbol{A}_i\boldsymbol{x}^* - b_i\right) = 0. \tag{3d}$$

Define slack variables $\boldsymbol{s}^* := \boldsymbol{b} - \boldsymbol{A}\boldsymbol{x}^*$, the inequality constraints become equalities $\boldsymbol{A}\boldsymbol{x}^* = \boldsymbol{b} - \boldsymbol{s}^*$. The KKT conditions can then be compactly written as,

$$\begin{bmatrix} \boldsymbol{Q} & \boldsymbol{A}^\mathsf{T} \\ \boldsymbol{A} & \boldsymbol{0} \end{bmatrix} \begin{bmatrix} \boldsymbol{x}^* \\ \boldsymbol{\lambda}^* \end{bmatrix} = \begin{bmatrix} -\boldsymbol{c} \\ \boldsymbol{b} - \boldsymbol{s}^* \end{bmatrix}, \tag{4}$$

where the inequality and complementarity conditions Eqs. (3c) and (3d) are implicitly encoded via $\boldsymbol{s}^* \geq \boldsymbol{0}$ and $s_i^*\lambda_i^* = 0$. In practice, we can partition the inequality constraints into *active* ones $\boldsymbol{A}_a\boldsymbol{x}^* = \boldsymbol{b}_a$ and *inactive* ones $\boldsymbol{A}_{\bar{a}}\boldsymbol{x}^* < \boldsymbol{b}_{\bar{a}}$, where the slack variables satisfy $\boldsymbol{s}_a^* = \boldsymbol{0}$ and $\boldsymbol{s}_{\bar{a}}^* > \boldsymbol{0}$.

**MPNNs for LCQPs** Representing LPs and QPs with MPNNs has been explored in prior work. For example, Chen et al. [2022] models LPs using a bipartite constraint-variable graph, and Chen et al. [2024a] extends this to QPs by adding edges between variable nodes. We adopt the setting of Chen et al. [2024a]. Given an LCQP instance $I$, we construct a graph $G(I)$ with constraint nodes $C(I)$ and variable nodes $V(I)$. Edges between $C(I)$ and $V(I)$ are defined by nonzero entries of $\boldsymbol{A}$ with weights $A_{cv}$, for $v \in V(I), c \in C(I)$; and edges between variables are defined by nonzero $Q_{vu}, v, u \in V(I)$. Node features are $\boldsymbol{H}_c := \text{reshape}(\boldsymbol{b}) \in \mathbb{R}^{m\times 1}$ for constraint nodes and $\boldsymbol{H}_v := \text{reshape}(\boldsymbol{c}) \in \mathbb{R}^{n\times 1}$ for variable nodes. MPNNs learn a vectorial representation of each node in a graph by aggregating information from its neighbors, i.e.,

$$\begin{aligned}
\boldsymbol{h}_c^{(t)} &:= \mathsf{UPD}_c^{(t)}\Big(\boldsymbol{h}_c^{(t-1)}, \sum_{v\in N(c)\cap V(I)} A_{cv}\boldsymbol{h}_v^{(t-1)}\Big) \\
\boldsymbol{h}_v^{(t)} &:= \mathsf{UPD}_v^{(t)}\Big(\boldsymbol{h}_v^{(t-1)}, \sum_{u\in N(v)\cap V(I)} Q_{uv}\boldsymbol{h}_u^{(t-1)}, \sum_{c\in N(v)\cap C(I)} A_{cv}\boldsymbol{h}_c^{(t)}\Big),
\end{aligned} \tag{5}$$

followed by a pooling function and a readout function to predict the objective,

$$\boldsymbol{z}_I := \mathsf{POOL}\Big(\sum_{v\in V(I)} \boldsymbol{h}_v^{(T)}, \sum_{c\in C(I)} \boldsymbol{h}_c^{(T)}\Big); \quad \text{obj}(I) := \mathsf{READOUT}(\boldsymbol{z}_I). \tag{6}$$

## 2 Principled data augmentation for LCQPs

We propose a set of principled transformations for LCQPs as data augmentation. Let $\mathcal{I}$ denote a set of LCQP instances. We consider transformations $T: \mathcal{I} \to \mathcal{I}$ that perturb the problem structure while allowing efficient computation of optimal solutions through simple linear-algebraic operations, without requiring a solver. Our core idea is to construct affine transformations of the KKT system Eq. (4) that yield valid KKT conditions for a new problem. We notice that applying a linear mapping $\boldsymbol{M}$ on Eq. (4) does not change the equality, i.e.,

$$\boldsymbol{M}\begin{bmatrix} \boldsymbol{Q} & \boldsymbol{A}^\mathsf{T} \\ \boldsymbol{A} & \boldsymbol{0} \end{bmatrix}\begin{bmatrix} \boldsymbol{x}^* \\ \boldsymbol{\lambda}^* \end{bmatrix} = \boldsymbol{M}\begin{bmatrix} -\boldsymbol{c} \\ \boldsymbol{b} - \boldsymbol{s}^* \end{bmatrix}$$

---

[1]We consider only feasible and bounded problems where $\boldsymbol{x}^*$ and $\boldsymbol{\lambda}^*$ are not both zeros. If they are, then $\boldsymbol{c} = \boldsymbol{0}$ and $\boldsymbol{b} \geq \boldsymbol{0}$, reducing the problem to an unconstrained QP with $\boldsymbol{Q} \succ \boldsymbol{0}$ and trivial solution $\boldsymbol{x}^* = \boldsymbol{0}$, which we exclude.

holds for all choices of transformation matrix $M$. To make the transformed problem valid, we need a $M^\intercal$ on the right and further require a right (pseudo) inverse $(M^\intercal)^\dagger$ with $M^\intercal (M^\intercal)^\dagger = I$. To ensure $(M^\intercal)^\dagger$ exists, the matrix $M$ needs to be full column rank. Now we have

$$M \begin{bmatrix} Q & A^\intercal \\ A & 0 \end{bmatrix} M^\intercal (M^\intercal)^\dagger \begin{bmatrix} x^* \\ \lambda^* \end{bmatrix} = M \begin{bmatrix} -c \\ b - s^* \end{bmatrix}.$$

Finally, we could add a bias term $B$ on both sides, and arrive at the general form,

$$\left( M \begin{bmatrix} Q & A^\intercal \\ A & 0 \end{bmatrix} M^\intercal + B \right) (M^\intercal)^\dagger \begin{bmatrix} x^* \\ \lambda^* \end{bmatrix} = M \begin{bmatrix} -c \\ b - s^* \end{bmatrix} + \beta. \tag{7}$$

Here, the matrices $M$ and $B$ define the transformation on the problem parameters $(Q, A, b, c)$, with $B$ chosen to satisfy $B (M^\intercal)^\dagger \begin{bmatrix} x^* \\ \lambda^* \end{bmatrix} = \beta$. The quantity $(M^\intercal)^\dagger \begin{bmatrix} x^* \\ \lambda^* \end{bmatrix}$, if it exists, recovers the optimal solution of the transformed problem. Thus, we can compute the new solutions using only matrix operations, without solving the transformed LCQP.

## 2.1 A framework for valid transformations

An ideal transformation should be expressive, valid, and computationally efficient. Here, *expressivity* refers to its ability to generate a wide range of problem instances from a given one, while *validity* ensures the transformed system remains a valid KKT system. We discuss these properties in detail below.

**Expressivity** Intuitively, a transformation $T$ is more expressive than another $T'$ if it maps an initial problem to a superset of the targets $T'$ can reach. We observe that the transformation in Eq. (7) has maximal expressivity. For example, setting $M = 0$, with proper $B$ and $\beta$ it can recover any target instance $I' = (Q', A', b', c')$ from any source $I$, regardless of dimensions. This flexibility relies on the bias term, without which the transformations are limited. For instance, a low-rank $A$ cannot map to a higher-rank $A'$. In practice, full expressivity is unnecessary, as our practical objective is not to span the entire space of QP instances, but to generate meaningful, diverse variants for data augmentation.

**Validity** Transformation following Eq. (7), under some conditions, can ensure that the transformed system remains a valid KKT system, which we explore below. We can write the matrices in block matrix form $M := \begin{bmatrix} M_{11} & M_{12} \\ M_{21} & M_{22} \end{bmatrix}$ and $B := \begin{bmatrix} B_{11} & B_{12} \\ B_{21} & B_{22} \end{bmatrix}$, such that dimensions for $A$ and $Q$ are matched. Using straightforward calculations, we have

$$\begin{bmatrix} T_{11} & T_{12} \\ T_{21} & T_{22} \end{bmatrix} = \begin{bmatrix} M_{11} & M_{12} \\ M_{21} & M_{22} \end{bmatrix} \begin{bmatrix} Q & A^\intercal \\ A & 0 \end{bmatrix} \begin{bmatrix} M_{11}^\intercal & M_{21}^\intercal \\ M_{12}^\intercal & M_{22}^\intercal \end{bmatrix} + \begin{bmatrix} B_{11} & B_{12} \\ B_{21} & B_{22,} \end{bmatrix}$$

$$T_{11} = M_{11} Q M_{11}^\intercal + M_{12} A M_{11}^\intercal + M_{11} A^\intercal M_{12}^\intercal + B_{11}$$
$$T_{12} = M_{11} Q M_{21}^\intercal + M_{12} A M_{21}^\intercal + M_{11} A^\intercal M_{22}^\intercal + B_{12}$$
$$T_{21} = M_{21} Q M_{11}^\intercal + M_{22} A M_{11}^\intercal + M_{21} A^\intercal M_{12}^\intercal + B_{21}$$
$$T_{22} = M_{21} Q M_{21}^\intercal + M_{22} A M_{21}^\intercal + M_{21} A^\intercal M_{22} + B_{22}.$$

So that the transformed equation Eq. (7) is a valid KKT form $\begin{bmatrix} Q' & A'^\intercal \\ A' & 0 \end{bmatrix} \begin{bmatrix} x'^* \\ \lambda'^* \end{bmatrix} = \begin{bmatrix} -c' \\ b' - s'^* \end{bmatrix}$, it requires $T_{12} = T_{21}^\intercal$, and we further require $T_{11}$ be another positive definite matrix, and $T_{22} = 0$ must hold.

For inequality constraints, the transformation matrix $M_{22}$ must satisfy certain conditions to enable efficient computation of the new solution. These conditions are not required for feasibility or optimality but reflect our goal of avoiding QP solving. To isolate $M_{22}$, we simplify the system by discarding the bias terms $B, \beta$, setting $M_{12}, M_{21}$ to zero, and fixing $M_{11} = I$. The resulting transformed KKT system is,

$$\begin{bmatrix} Q & A^\intercal M_{22}^\intercal \\ M_{22} A & 0 \end{bmatrix} \begin{bmatrix} I & 0 \\ 0 & (M_{22}^\intercal)^\dagger \end{bmatrix} \begin{bmatrix} x^* \\ \lambda^* \end{bmatrix} = \begin{bmatrix} I & 0 \\ 0 & M_{22} \end{bmatrix} \begin{bmatrix} -c \\ b - s^* \end{bmatrix}. \tag{8}$$

This system indicates that the primal solution $x^*$ remains unchanged, while the dual solution is mapped to $(M_{22}^\mathsf{T})^\dagger \lambda^*$. The following proposition establishes the conditions under which this transformation preserves the solution and avoids re-solving the LCQPs; see Appendix D.1 for a proof.

**Proposition 2.1.** *Let $I := (Q, A, b, c)$ be a LCQP instance with optimal primal-dual solution $x^*, \lambda^*$. Consider a transformation defined by $T(I) := (Q, M_{22}A, M_{22}b, c)$. Then the transformed problem preserves the primal solution $x^*$ if and only if $M_{22}$ takes the block form $\begin{bmatrix} N_{11} & N_{12} \\ N_{21} & N_{22} \end{bmatrix}$ to match the dimensions of active and inactive constraints, where $N_{11}$ has full row rank equal to the number of active constraints, $N_{12}s_{\bar{a}}^* = 0$, $N_{22}s_{\bar{a}}^* \geq 0$.*

In summary, the conditions on $M, B, \beta$ for the validity of the new problem is as follows,

$$
\begin{aligned}
M_{11}QM_{11}^\mathsf{T} + M_{12}AM_{11}^\mathsf{T} + M_{11}A^\mathsf{T}M_{12}^\mathsf{T} + B_{11} &\succ 0 \\
M_{21}QM_{21}^\mathsf{T} + M_{22}AM_{21}^\mathsf{T} + M_{21}A^\mathsf{T}M_{22} + B_{22} &= 0 \\
B_{12} &= B_{21}^\mathsf{T} \\
B(M^\mathsf{T})^\dagger \begin{bmatrix} x^* \\ \lambda^* \end{bmatrix} &= \beta, \text{ and} \\
M_{22} \text{ satisfies Proposition 2.1.}
\end{aligned}
\tag{9}
$$

**Computational efficiency** In general, satisfying the full transformation structure in Eq. (9) is challenging, particularly when arbitrary $M, B$ and $\beta$ are involved. However, we notice the multiplicative term $M$ and bias terms $B, \beta$ can be decoupled. That is, we can design transformations with the bias terms $B, \beta$, e.g. Appendix C.2, but we mainly focus on dropping them, setting $M_{12}$ and $M_{21}$ to zero matrices, and investigate the design space of $M_{11}$ and $M_{22}$. We will abbreviate the subscripts of $M_{ii}$, and we target at designing transformations of the form

$$
\begin{bmatrix} M_1 & 0 \\ 0 & M_2 \end{bmatrix} \begin{bmatrix} Q & A^\mathsf{T} \\ A & 0 \end{bmatrix} \begin{bmatrix} M_1^\mathsf{T} & 0 \\ 0 & M_2^\mathsf{T} \end{bmatrix} \begin{bmatrix} (M_1^\mathsf{T})^\dagger & 0 \\ 0 & (M_2^\mathsf{T})^\dagger \end{bmatrix} \begin{bmatrix} x^* \\ \lambda^* \end{bmatrix} = \begin{bmatrix} M_1 & 0 \\ 0 & M_2 \end{bmatrix} \begin{bmatrix} -c \\ b - s^* \end{bmatrix}. \tag{10}
$$

We notice the commutativity and the decoupled nature,

$$
\begin{bmatrix} M_1 & 0 \\ 0 & M_2 \end{bmatrix} = \begin{bmatrix} I & 0 \\ 0 & M_2 \end{bmatrix} \begin{bmatrix} M_1 & 0 \\ 0 & I \end{bmatrix} = \begin{bmatrix} M_1 & 0 \\ 0 & I \end{bmatrix} \begin{bmatrix} I & 0 \\ 0 & M_2 \end{bmatrix},
$$

which enables us to design $M_1, M_2$ separately, and merge them later.

The design space in Eq. (10) is flexible but constrained by the need to compute pseudo-inverses. For $M_1 \in \mathbb{R}^{n' \times n}$, there are three cases: (i) $n' = n$ linear reparameterization, (ii) $n' < n$, dropping variables, and (iii) $n' > n$ adding variables. While (ii) is often ill-posed since $M_1^\mathsf{T}$ does not have full column rank and lacks a right inverse, some special cases are still feasible, as we will show. More broadly, computing $(M_1^\mathsf{T})^\dagger = M_1 (M_1^\mathsf{T}M_1)^{-1}$ requires full column rank and typically costs $\mathcal{O}(n^3)$. To improve efficiency, we focus on structured matrices, for example, diagonal matrices, as a particular case, enable both efficient inverse calculation and generation in $\mathcal{O}(n)$ time. These considerations motivate the use of diagonal or structured $M_1$ and $M_2$ for scalable data augmentation. To formalize efficiency, we introduce two notions of efficiently computable transformations. The first covers transformations whose solutions can be computed in linear time.

**Definition 2.2** (Efficiently recoverable transformation). *For an LCQP instance $I := (Q, A, b, c) \in \mathcal{I}$ and its primal-dual solutions $x^*, \lambda^*$, a transform $T \in \mathcal{T}$ is* efficiently recoverable, *if the new solutions of the transformed instance $\mathcal{I}' = T(\mathcal{I})$ can be obtained within linear time $\mathcal{O}(n)$.*

The second one, motivated by unsupervised settings, tightens this by requiring that the transformation be independent of the original solution and focuses on generating structurally consistent instances rather than solving them.

**Definition 2.3** (Solution-independent transformation). *Given a LCQP instance $I := (Q, A, b, c) \in \mathcal{I}$, a transformation $T \in \mathcal{T}$ is* solution-independent *if the solution of the transformed problem can be obtained without a solver, and the transformation parameters do not depend on the original optimal solutions $x^*, \lambda^*$.*

## 2.2 Example transformations

In the following, we discuss various transformations that fit into the above framework.

**Removing idle variables**  As discussed above, when $M_1$ does not have full column rank, the pseudo inverse $(M_1^\mathsf{T})^\dagger$ typically does not exist, making such transformations ill-posed. However, there is a special case in which we can safely remove a variable, specifically, when it is idle because its optimal value is zero, resulting in the following proposition.

**Proposition 2.4.** *Let $I := (Q, A, b, c)$ be an LCQP instance with primal-dual solution $x^*, \lambda^*$. Then a variable $x'$ can be removed from the problem without affecting the optimal values of the remaining variables if and only if $x'^* = 0$.*

**Removing inactive constraints**  Like the variable removal case above, a constraint can be removed under certain conditions by introducing a wide identity matrix $M_2$ that selectively excludes the corresponding row.

**Proposition 2.5.** *Let $I := (Q, A, b, c)$ be an LCQP instance with optimal primal-dual solution $x^*, \lambda^*$. Then a constraint of the form $a'^\mathsf{T} x^* \leq b'$ can be removed from the problem without affecting the optimal solution if and only if it is strictly inactive, i.e., $a'^\mathsf{T} x^* < b'$.*

The variable and constraint removal transformations are *efficiently recoverable*, as the remaining solutions are unchanged and need no recomputation. However, they are not *solution-independent*, since identifying removable components requires access to the primal solution. Moreover, we apply a heuristic on problem instances to select inactive constraints, as described in Appendix C.3.

**Scaling variable coefficients**  A natural class of transformations involves scaling the coefficients associated with individual variables. Specifically, scaling the $j$-th column of $A$ and the $j$-th entry of $c$ by a nonzero scalar $\alpha_j$, while updating $Q_{ij}$ by $\alpha_i \alpha_j$, i.e., $T(I) := (M_1 Q M_1, A M_1^\mathsf{T}, b, M_1 c)$ with a diagonal $M_1$ preserves the structure of the QP. Under this transformation, the optimal value remains unchanged, and the solution $x_j^*$ is rescaled by $1/\alpha_j$. This transformation is *efficiently recoverable*, as the new solution can be obtained directly from the original in $\mathcal{O}(n)$ time. Moreover, it is also *solution-independent*, as it does not require access to the original solution, but only knowledge of how to compute the new one.

**Adding variables**  When $n' > n$, the transformation effectively adds new variables to the problem and linearly combines existing ones. Without loss of generality, we consider adding a single new variable by choosing a transformation matrix of the form $M_1 := \begin{bmatrix} I \\ q^\mathsf{T} \end{bmatrix}$, with $q \in \mathbb{R}^n$ being an arbitrary vector. This yields a new positive definite quadratic matrix $\begin{bmatrix} Q & Qq \\ q^\mathsf{T}Q & q^\mathsf{T}Qq \end{bmatrix}$. We can find the pseudo inverse $(M_1^\mathsf{T})^\dagger := M_1 (M_1^\mathsf{T} M_1)^{-1} = \begin{bmatrix} I \\ q^\mathsf{T} \end{bmatrix} (I + qq^\mathsf{T})^{-1}$, which can be computed with Sherman-Morrison formula [Shermen and Morrison, 1949]. In this case, the primal solution of the original variables does not remain the same. Interestingly, due to the structure of $M_1$, another valid pseudo-inverse is $(M_1^\mathsf{T})^\dagger = \begin{bmatrix} I \\ 0^\mathsf{T} \end{bmatrix}$, which is a special case indicating that the added variable has zero contribution to the solution. This corresponds to the reverse of the variable removal transformation discussed above.

**Proposition 2.6.** *Let $I = (Q, A, b, c)$ be an LCQP instance with optimal solution $x^*, \lambda^*$. Define the transformation $T(I) := (M_1 Q M_1, A M_1^\mathsf{T}, b, M_1 c)$, where $M_1 := \begin{bmatrix} I \\ q^\mathsf{T} \end{bmatrix}$. Then the transformed problem has optimal primal solution $(x^*, x'^*)$ if and only if the new variable $x'^* = 0$.*

This transformation is both *efficiently recoverable* and *solution-independent*. Moreover, there also exist other implementations of variable addition in the form of Eq. (7), with a bias term. Please refer to Appendix C.4.

**Scaling constraints**  Similar to the variable scaling transformation above, we can scale the constraint coefficients by fixing $M_1 = I$ and letting $M_2$ be a square diagonal matrix. We assume all diagonal

entries of $M_2$ are positive. If any entry is zero, the transformation reduces to inactive constraint removal above, or an active constraint is removed, and the problem will be relaxed. If any entry is negative, the corresponding inequality direction will be flipped, and the problem's solution may change. Specifically, the transformation would be $T(I) := (Q, M_2 A, M_2 b, c)$.

Under this transformation, the new dual variables are given by $\lambda'^* := M_2^{-1} \lambda^*$, which can be computed in linear time since $M_2$ is a diagonal matrix. The primal solution remains unchanged. This transformation is both *efficiently recoverable* and *solution-independent*.

In Appendix C, we outline additional transformations, including constraint addition Appendix C.1.

### 2.3 Using data augmentations

Having introduced efficient data augmentation methods for LCQPs and LPs, we now describe how they can be integrated into different training pipelines. Our target task is graph regression to predict the objective value from a graph representation. Depending on the setting, we can either (1) use solution information to generate supervised labels for related problems or (2) apply solution-independent augmentations for contrastive pretraining without solutions.

**Supervised learning**   Given a training set and a set of augmentation methods, we dynamically generate additional training instances during optimization. We randomly apply a selected augmentation or a combination of augmentations at each iteration, and train the MPNN using the supervised loss on the predicted objective value.

**Contrastive pretraining**   We perform self-supervised contrastive learning on the entire dataset without touching the solutions of the problems, using the NT-Xent loss [Chen et al., 2020]. Specifically, for a mini-batch of $N$ data instances, we generate two augmented views of each instance using solution-independent transformations, resulting in $2N$ data instances. We consider the two views of the same instance $(i, j)$ as a positive pair, and the other $2(N-1)$ samples as negative. The similarity between embeddings is measured by 2-norm-normalized cosine similarity $\mathsf{sim}(u, v) := \frac{u^\top v}{\|u\|\|v\|}$. For each instance $i$ and its positive sample $j$, we have the NT-Xent loss on the pooled representations $z_i, z_j$ from Eq. (6) as

$$-\frac{1}{N} \sum_{i=1}^{N} \log \frac{\exp\left(\mathsf{sim}(z_i, z_j)/\tau\right)}{\sum_{k=1}^{2N} \mathbb{I}(k \neq i) \exp\left(\mathsf{sim}(z_i, z_k)/\tau\right)}, \tag{11}$$

where $\tau > 0$ is a temperature hyperparameter.

## 3   Experimental setup and results

To empirically validate the effectiveness of our data augmentations, we conduct a series of experiments, answering the following research questions.[2]

**Q1** Do our augmentations improve supervised learning, especially under data scarcity?
**Q2** Are the augmentations effective in contrastive pretraining followed by supervised finetuning?
**Q3** Does the pretrained model enhance generalization on out-of-distribution (OOD) or larger datasets?
**Q4** What is the practical computational overhead of the augmentations?

As LPs are a special case of QPs, we evaluate them separately. We generate $10\,000$ instances for each dataset, each with 100 variables and 100 inequality constraints, and split the data into training, validation, and test sets with $8 : 1 : 1$ ratio. To study performance under data scarcity, we partition the training set into multiple disjoint subsets in each run, each containing either 10% or 20% of the full training data. Models are trained independently on each subset, and results are aggregated across all partitions. Hyperparameters are tuned using supervised training on the full training set and fixed across all methods. Specifically, we use a 6-layer MPNN followed by a 3-layer MLP with 192 hidden dimensions and GraphNorm [Cai et al., 2021]. All experiments are conducted on a single NVIDIA L40S GPU. We evaluate performance using the mean relative objective error (in percentage) over the test set, defined as $1/|\mathcal{D}| \sum_{I \in \mathcal{D}} |(\mathrm{obj}(I) - \mathrm{obj}^*(I))/\mathrm{obj}^*(I)| \cdot 100\%$, where $\mathcal{D}$ denotes the set of

---

[2]The repository of our source code can be accessed at `https://github.com/chendiqian/Data-Augmentation-for-Learning-to-Optimize`.

instances, $\text{obj}^*(I)$ is the optimal objective value, and $\text{obj}(I)$ is the predicted objective value. We repeat the experiments five times with different seeds and report the mean and standard deviation.

**Supervised learning**     To address **Q1**, we investigate the impact of data augmentation on LPs and QPs in a supervised learning setting. We evaluate performance on synthetically generated datasets following the procedure described in Appendix G. Models are trained with a batch size of 32 for up to 2000 epochs, with early stopping after 200 epochs of patience. We evaluate performance under data scarcity by training models on subsets containing 10%, 20%, and 100% of the training data. As baselines, we apply the augmentations proposed by You et al. [2020]: node dropping, edge perturbation, and feature masking. In addition, we evaluate each of our proposed augmentations separately and in combination. Hyperparameter details for all augmentations are provided in Appendix F. Results, shown in Table 1, reveal that the augmentations from You et al. [2020] fail to improve performance consistently and can even be detrimental compared to training without augmentation. In contrast, all of our proposed methods consistently yield better performance, and combining augmentations further amplifies the improvement up to 62.6% on LP with 20% of training data, aligning with empirical evidence observed in You et al. [2020].

Table 1: Supervised learning performance on LP/QP datasets with and without data augmentation under different levels of data scarcity. The best-performing method is colored in green, the second-best in blue, and third in orange. Our proposed augmentations consistently improve performance.

| Augmentation | LP | | | QP | | |
|---|---|---|---|---|---|---|
| | 10% | 20% | 100% | 10% | 20% | 100% |
| None | $8.539_{\pm 0.206}$ | $6.149_{\pm 0.153}$ | $2.784_{\pm 0.081}$ | $5.304_{\pm 0.229}$ | $3.567_{\pm 0.034}$ | $1.240_{\pm 0.088}$ |
| Drop node | $8.618_{\pm 0.152}$ | $7.131_{\pm 0.072}$ | $4.654_{\pm 0.047}$ | $6.355_{\pm 0.254}$ | $4.867_{\pm 0.126}$ | $2.594_{\pm 0.087}$ |
| Mask node | $8.979_{\pm 0.153}$ | $7.591_{\pm 0.117}$ | $3.216_{\pm 0.149}$ | $5.865_{\pm 0.414}$ | $4.005_{\pm 0.161}$ | $1.347_{\pm 0.067}$ |
| Flip edge | $7.794_{\pm 0.115}$ | $6.503_{\pm 0.125}$ | $4.358_{\pm 0.081}$ | $5.485_{\pm 0.133}$ | $4.361_{\pm 0.138}$ | $2.561_{\pm 0.091}$ |
| Drop vars. | $4.996_{\pm 0.094}$ | $3.484_{\pm 0.014}$ | $1.484_{\pm 0.070}$ | $4.232_{\pm 0.195}$ | $2.374_{\pm 0.085}$ | $0.835_{\pm 0.042}$ |
| Drop cons. | $5.563_{\pm 0.097}$ | $3.691_{\pm 0.087}$ | $1.656_{\pm 0.057}$ | $3.407_{\pm 0.099}$ | $2.104_{\pm 0.073}$ | $0.821_{\pm 0.063}$ |
| Scale cons. | $5.491_{\pm 0.093}$ | $3.872_{\pm 0.136}$ | $1.612_{\pm 0.121}$ | $3.909_{\pm 0.142}$ | $2.448_{\pm 0.033}$ | $0.722_{\pm 0.058}$ |
| Scale vars. | $4.682_{\pm 0.162}$ | $3.208_{\pm 0.116}$ | $1.241_{\pm 0.052}$ | $3.790_{\pm 0.013}$ | $2.468_{\pm 0.124}$ | $0.649_{\pm 0.040}$ |
| Add cons. | $6.245_{\pm 0.118}$ | $4.299_{\pm 0.048}$ | $1.878_{\pm 0.059}$ | $3.731_{\pm 0.065}$ | $2.304_{\pm 0.042}$ | $0.814_{\pm 0.052}$ |
| Add vars. | $6.565_{\pm 0.118}$ | $4.696_{\pm 0.101}$ | $2.188_{\pm 0.102}$ | $4.475_{\pm 0.276}$ | $2.885_{\pm 0.045}$ | $1.021_{\pm 0.055}$ |
| Combo | $3.465_{\pm 0.045}$ | $2.300_{\pm 0.073}$ | $1.051_{\pm 0.021}$ | $2.434_{\pm 0.061}$ | $1.532_{\pm 0.033}$ | $0.542_{\pm 0.013}$ |

**Contrastive pretraining**     To address **Q2**, we evaluate whether contrastive pretraining can improve supervised fine-tuning, potentially under data scarcity. We adopt a semi-supervised setting: a small subset $(10\%, 20\%, 100\%)$ of the training data is labeled, while the whole training set is available as unlabeled data. During pretraining, we use the complete unlabeled training set and train only an MPNN backbone without a prediction head, following the deployment described in Section 2.3. We pretrain for 800 epochs with a batch size of 128, and set $\tau = 0.1$. To assess pretraining quality and pick the best set of hyperparameters, we use linear probing [Veličković et al., 2018], training only a linear regression layer on top of a frozen MPNN to efficiently evaluate feature quality. For finetuning, we follow Zeng and Xie [2021], attaching an MLP head and jointly training it with the MPNN using supervised regression loss, essentially the same setup as supervised learning, but initialized from a pretrained model.

As baselines, we consider several graph contrastive learning methods. GraphCL [You et al., 2020] generates views via node dropping, edge perturbation, and feature masking. GCC [Qiu et al., 2020] samples random walk subgraphs. IGSD [Zhang et al., 2023] uses graph diffusion [Gasteiger et al., 2019] combined with model distillation. MVGRL [Hassani and Khasahmadi, 2020] also employs diffusion-based views but performs contrastive learning at the graph-node level. Additionally, we include mutual information maximization methods such as InfoGraph [Sun et al., 2019] and DGI [Veličković et al., 2018]. Beyond contrastive methods, we evaluate the generative SSL method GAE [Kipf and Welling, 2016], which reconstructs graph edge weights.

As shown in Table 2, GraphCL and GCC pretraining can improve performance in some cases but do not consistently yield better results. Other baselines even degrade performance. In contrast, our pretraining methods substantially improve, reducing the objective gap by 59.4% on LP and 54.1% on QP with only 10% of the training data. This highlights the effectiveness of our data augmentations, which are specifically tailored for optimization problem instances and significantly enhance fine-tuning.

Table 2: Pretrained-finetuned model performance on LP/QP datasets under different levels of data scarcity. Our pretraining consistently improves performance and outperforms the baselines.

| Pretraining | LP | | | QP | | |
|---|---|---|---|---|---|---|
| | 10% | 20% | 100% | 10% | 20% | 100% |
| None | $8.539_{\pm 0.206}$ | $6.149_{\pm 0.153}$ | $2.784_{\pm 0.081}$ | $5.304_{\pm 0.229}$ | $3.567_{\pm 0.034}$ | $1.240_{\pm 0.088}$ |
| GraphCL | $9.694_{\pm 1.656}$ | $6.033_{\pm 0.653}$ | $2.613_{\pm 0.266}$ | $6.024_{\pm 0.859}$ | $3.865_{\pm 0.296}$ | $1.253_{\pm 0.114}$ |
| GCC | $8.248_{\pm 0.916}$ | $5.761_{\pm 0.186}$ | $2.686_{\pm 0.201}$ | $11.344_{\pm 0.198}$ | $6.356_{\pm 0.964}$ | $1.472_{\pm 0.076}$ |
| IGSD | $18.097_{\pm 2.276}$ | $7.631_{\pm 0.794}$ | $3.082_{\pm 0.269}$ | $12.799_{\pm 1.485}$ | $6.306_{\pm 0.628}$ | $1.302_{\pm 0.094}$ |
| MVGRL | $20.392_{\pm 2.476}$ | $9.072_{\pm 1.854}$ | $2.852_{\pm 0.112}$ | $8.440_{\pm 0.981}$ | $4.932_{\pm 1.083}$ | $1.343_{\pm 0.063}$ |
| InfoGraph | $18.338_{\pm 3.285}$ | $7.464_{\pm 0.818}$ | $2.956_{\pm 0.179}$ | $10.306_{\pm 0.189}$ | $6.246_{\pm 0.180}$ | $1.409_{\pm 0.132}$ |
| DGI | $19.661_{\pm 4.468}$ | $9.671_{\pm 2.764}$ | $3.156_{\pm 0.188}$ | $10.014_{\pm 0.528}$ | $7.223_{\pm 0.142}$ | $1.425_{\pm 0.109}$ |
| GAE | $9.082_{\pm 0.901}$ | $6.032_{\pm 0.241}$ | $3.434_{\pm 0.255}$ | $5.848_{\pm 0.181}$ | $3.759_{\pm 0.158}$ | $1.381_{\pm 0.027}$ |
| Ours | $3.472_{\pm 0.086}$ | $2.794_{\pm 0.049}$ | $1.588_{\pm 0.056}$ | $3.791_{\pm 0.097}$ | $2.427_{\pm 0.083}$ | $0.926_{\pm 0.031}$ |

**Generalization** To address **Q3**, we compare the performance of models trained from scratch versus models initialized with contrastive pretraining and then finetuned. For LPs, we generate four types of relaxed LP instances derived from MILPs: Set Cover (SC), Maximum Independent Set (MIS), Combinatorial Auction (CA), and Capacitated Facility Location (CFL), following Gasse et al. [2019]. For QPs, we generate instances of soft-margin SVM, Markowitz portfolio optimization, and LASSO regression following Jung et al. [2022]. If possible, problem sizes and densities are kept similar to the pretraining datasets; see more details in Appendix G. As shown in Tables 3 and 4, pretrained models outperform models trained from scratch in almost all cases, demonstrating strong transferability to OOD tasks. The evaluation on larger datasets can be found in Appendix E.2.

Table 3: Generalization performance of contrastive pretrained MPNNs on OOD LP instances.

| Family | Pretrained | Ratio | | |
|---|---|---|---|---|
| | | 10% | 20% | 100% |
| SC | No | $5.297_{\pm 0.787}$ | $2.531_{\pm 0.861}$ | $0.752_{\pm 0.042}$ |
| | CL | $1.965_{\pm 0.206}$ | $1.349_{\pm 0.161}$ | $0.662_{\pm 0.055}$ |
| MIS | No | $0.674_{\pm 0.016}$ | $0.465_{\pm 0.022}$ | $0.203_{\pm 0.006}$ |
| | Yes | $0.722_{\pm 0.056}$ | $0.421_{\pm 0.047}$ | $0.177_{\pm 0.020}$ |
| CA | No | $2.381_{\pm 0.048}$ | $1.803_{\pm 0.027}$ | $0.906_{\pm 0.025}$ |
| | Yes | $2.303_{\pm 0.184}$ | $1.713_{\pm 0.138}$ | $0.753_{\pm 0.055}$ |
| CFL | No | $0.401_{\pm 0.011}$ | $0.255_{\pm 0.024}$ | $0.059_{\pm 0.006}$ |
| | CL | $\mathbf{0.367}_{\pm 0.047}$ | $0.217_{\pm 0.018}$ | $0.064_{\pm 0.008}$ |

Table 4: Generalization performance of contrastive pretrained MPNNs on OOD QP instances.

| Family | Pretrained | Ratio | | |
|---|---|---|---|---|
| | | 10% | 20% | 100% |
| SVM | No | $0.191_{\pm 0.006}$ | $0.109_{\pm 0.007}$ | $0.027_{\pm 0.004}$ |
| | Yes | $0.141_{\pm 0.017}$ | $0.077_{\pm 0.009}$ | $0.024_{\pm 0.004}$ |
| Portfolio | No | $3.766_{\pm 0.198}$ | $1.841_{\pm 0.162}$ | $0.402_{\pm 0.005}$ |
| | Yes | $3.331_{\pm 0.376}$ | $1.637_{\pm 0.013}$ | $0.353_{\pm 0.014}$ |
| LASSO | No | $5.405_{\pm 0.018}$ | $4.083_{\pm 0.025}$ | $2.159_{\pm 0.493}$ |
| | Yes | $5.169_{\pm 0.171}$ | $3.726_{\pm 0.267}$ | $1.178_{\pm 0.097}$ |

Regarding **Q4**, see Appendix E.1 for results and discussion.

## 4 Conclusion

We introduced a principled framework for data augmentation in learning to optimize over linear and quadratic programming. By leveraging affine transformations of the KKT system, we designed a family of expressive, solution-preserving, and computationally efficient transformations. Our method allows for augmentations that either admit exact solution recovery or preserve key structural properties without requiring access to the original solutions, making them suitable for supervised and contrastive learning. Extensive experiments show that these augmentations consistently improve performance under data scarcity, generalize to larger and out-of-distribution problems, and outperform existing graph augmentation baselines. This work highlights the benefits of optimization-aware augmentation strategies and opens new directions for robust, scalable L2O under limited supervision.

## Acknowledgements

Christopher Morris and Chendi Qian are partially funded by a DFG Emmy Noether grant (468502433) and RWTH Junior Principal Investigator Fellowship under Germany's Excellence Strategy. We thank Erik Müller for crafting the figures.

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

# A  Limitations

While our proposed data augmentation framework for QP and LP instances is efficient and principled, several limitations remain. First, some transformations, e.g., Appendix C.2 and Section 2.2, require access to the optimal primal-dual solution of the original problem, which may not always be available in practice. Second, our framework assumes convexity (i.e., $Q \succ 0$) and problem feasibility, and does not directly extend to non-convex or infeasible cases. Additionally, non-linear quadratic programming is beyond our scope. Also, we train and evaluate solely on synthetic QP and LP instances because real-world datasets are scarce and heterogeneous. Benchmarks such as MIPLIB [Gleixner et al., 2021] and QPLIB [Furini et al., 2019] contain too few samples and exhibit significant variability in problem size and structure, making them unsuitable for training deep models. Finally, while our transformation framework is mathematically expressive, we restrict it to a small subset of efficient, feasible augmentations in practice, leaving its full expressive power unexplored.

# B  Broader impact

This work introduces principled data augmentation methods for learning-based solvers in convex optimization, focusing on predicting objective values and solutions for LP/QP problems. These methods can also replace heuristics, such as strong branching in mixed-integer optimization. Our approach may benefit logistics, finance, and scientific computing applications by enabling more robust and data-efficient learning. As a foundational contribution, this work does not raise concerns about privacy, security, or fairness. It does not involve sensitive data or end-user interaction, and the proposed techniques are general-purpose and not tied to specific domains. Nevertheless, when applied to real-world optimization systems, care should be taken to ensure reliability and fairness.

# C  Additional transformations

Here, we outline additional transformations, omitted from the main paper for space reasons.

## C.1  Adding constraints

Analogous to adding variables (see Section 2.2), we can augment a problem instance by introducing additional constraints. This is done by extending the constraint matrix using a transformation of the form $M_2 := \begin{bmatrix} I \\ m^\mathsf{T} \end{bmatrix}$, where $m \in \mathbb{R}^m_{\geq 0}$ is a non-negative vector. This effectively appends a new constraint that is a convex combination of the existing ones. Notably, this transformation, along with the scaling constraints in Section 2.2, satisfies the condition derived in Eq. (13). In unsupervised settings where neither the primal-dual solutions nor the active constraints are known, we store all constraints and add a new one that is linearly combined with them and does not affect the solution. The added constraint remains inactive, corresponding to a zero dual variable, transforming both *efficiently recoverable* and *solution-independent*.

**Proposition C.1.** *Let $I = (Q, A, b, c)$ be an LCQP instance with optimal solution $(x^*, \lambda^*)$. We define our transformation as $T(I) := (Q, M_2 A, M_2 b, c)$, where $M_2 := \begin{bmatrix} I \\ m^\mathsf{T} \end{bmatrix}$. Then the transformed instance has an optimal solution $x^*, [\lambda^*, \lambda'^*]$ if and only if the new dual variable $\lambda'^* = 0$.*

## C.2  Biasing the problem

We introduce a data augmentation strategy that leverages nontrivial bias terms while fixing $M$ to the identity. Specifically,

$$\left( \begin{bmatrix} Q & A^\mathsf{T} \\ A & 0 \end{bmatrix} + B \right) \begin{bmatrix} x^* \\ \lambda^* \end{bmatrix} = \begin{bmatrix} -c \\ b - s^* \end{bmatrix} + \beta. \tag{12}$$

Consider the transformation in Eq. (12), where we fix $M := I$ and design a suitable bias matrix $B$. To satisfy the conditions in Eq. (9), we construct $B_{11} := RR^\mathsf{T}$, where $R \in \mathbb{R}^{n \times k}$ is a random matrix, ensuring that the resulting $Q$ remains positive definite. We then set $B_{21} = B_{12}^\mathsf{T} \in$

$\mathbb{R}^{m \times n}$ as a random matrix, and $\boldsymbol{B}_{22} := \boldsymbol{0}$. Accordingly, we must have $\boldsymbol{\beta} := \begin{bmatrix} \boldsymbol{B}_{11} & \boldsymbol{B}_{12} \\ \boldsymbol{B}_{21} & \boldsymbol{0} \end{bmatrix} \begin{bmatrix} \boldsymbol{x}^* \\ \boldsymbol{\lambda}^* \end{bmatrix}$, ensuring the transformed KKT system is satisfied. This yields the transformed instance $T(I) := (\boldsymbol{Q} + \boldsymbol{B}_{11}, \boldsymbol{A} + \boldsymbol{B}_{21}, \boldsymbol{b} + \boldsymbol{B}_{21}\boldsymbol{x}^*, \boldsymbol{c} - \boldsymbol{B}_{11}\boldsymbol{x}^* + \boldsymbol{A}^\mathsf{T}\boldsymbol{\lambda}^*)$ under which the original primal and dual solutions remain valid; therefore, it is an efficiently recoverable transformation. However, this transformation is not solution-independent, since it requires access to $\boldsymbol{x}^*, \boldsymbol{\lambda}^*$, which is an intrinsic drawback of such biasing transformations.

### C.3 Finding inactive constraints

We introduce the following heuristics. For all inequality constraints in a given problem, including the variable bounds, e.g., $x_i \geq 0$, we calculate a score $h_i := \boldsymbol{a}_i^\mathsf{T}\boldsymbol{x} + b_i$. The constraints with lower scores are more likely to be active. See Appendix C.3 for an illustration. Given the number of constraints $m$

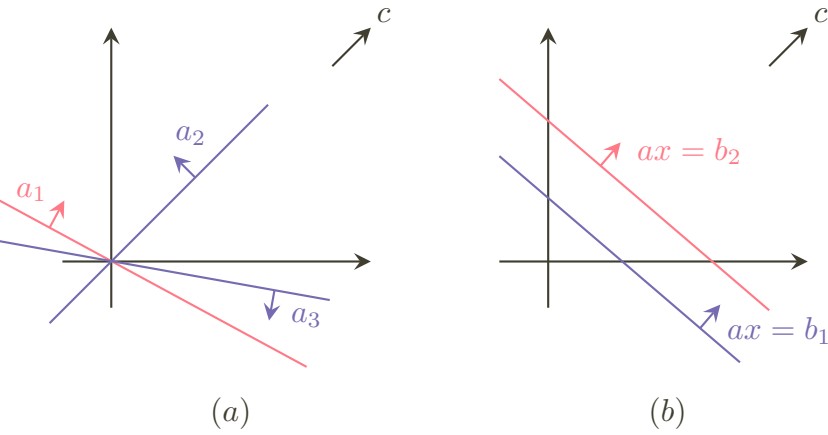

Figure 2: Illustration of our heuristics. (a). Smaller $\boldsymbol{a}^\mathsf{T}\boldsymbol{c}$ is more likely to be inactive. (b). Lower $b$ is more likely to be inactive.

and the number of variables $n$, if $m > n$ (due to variable bounds constraints) and the constraints are not degenerate, $n$ constraints will be active. Therefore, we pick the $m - n$ constraints with the largest $h_i$ as heuristic inactive constraints. Then this could be plugged into the removing constraint transformation.

We evaluate the effectiveness on our datasets, by calculating the accuracy given ground truth inactive constraint set $\bar{\mathcal{A}}_{\text{gt}}$ and heuristic one $\bar{\mathcal{A}}_{\text{heu}}$,

$$\text{acc} := \frac{|\bar{\mathcal{A}}_{\text{gt}} \cap \bar{\mathcal{A}}_{\text{heu}}|}{\bar{\mathcal{A}}_{\text{heu}}}$$

over all the instances; see Table 5 for a summary of our results.

Table 5: Accuracy summary.

| | LP | | | | QP | | | |
|---|---|---|---|---|---|---|---|---|
| Size | 100 | 150 | 200 | 250 | 100 | 150 | 200 | 250 |
| Acc. (%) | 88.5±2.9 | 90.1±3.2 | 89.1±2.8 | 88.3±3.6 | 91.8±3.4 | 92.9±2.7 | 91.6±2.1 | 91.4±2.4 |

The heuristic's performance is satisfactory, with a low false-positive rate. Although it is generally designed for LPs, it is also effective for QPs.

### C.4 Adding variable

Besides Proposition 2.6, we derive another efficient method to add a new variable.

**Proposition C.2.** *Given an LCQP instance $I := (Q, A, b, c)$ and its optimal primal and dual solution $x^*, \lambda^*$, the data augmentation $T(I) := \left( \begin{bmatrix} Q & 0 \\ 0^\intercal & q \end{bmatrix}, [A \quad a], b, \begin{bmatrix} c \\ c' \end{bmatrix} \right)$, where $q > 0$, $a \in \mathbb{R}^m$ is arbitrary, $c' := -a^\intercal \lambda^*$, allows for an extra variable $x'$ without affecting the original optimal solution, if and only if the new variable $x'^* = 0$.*

Such data transformation construction is efficient and valid. The only drawback is that it is not solution-independent. Therefore, we provide an improved augmentation that introduces a new constraint.

**Corollary C.2.1.** *Given a QP instance $I := (Q, A, b, c)$ and its optimal primal and dual solution $x^*, \lambda^*$, the data augmentation $T(I) := \left( \begin{bmatrix} Q & 0 \\ 0^\intercal & q \end{bmatrix}, \begin{bmatrix} A & a \\ 0^\intercal & 1 \end{bmatrix}, \begin{bmatrix} b \\ 0 \end{bmatrix}, \begin{bmatrix} c \\ c' \end{bmatrix} \right)$, where $q > 0$, $a \in \mathbb{R}^m$, $c' \in \mathbb{R}$ is arbitrary, allows for an extra variable $x'$ an extra dual variable $\lambda'$ without affecting the original optimal solution if and only if the new primal variable $x'^* = 0$.*

**Proposition C.3.** *There exists a set of matrices $\{M, B, \beta\}$, such that the design in Corollary C.2.1 can be realized in the form of Eq. (7).*

# D   Omitted proofs

Here, we outline missing proofs from the main paper.

## D.1   Proof for Proposition 2.1

*Proof.* The rank condition of $N_{11}$ is straightforward. Calculating the (pseudo) inverse $\begin{bmatrix} N_{11}^\intercal & N_{21}^\intercal \\ N_{12}^\intercal & N_{22}^\intercal \end{bmatrix}^\dagger$ requires $N_{11}$ or $N_{22}$ being invertible. From the KKT condition Eq. (3d), we know $\lambda_{\bar{a}}^* = 0$. So we don't have to consider $N_{22}^{\intercal\,-1}$ or its existence; however, $N_{11}$ must be invertible, otherwise we cannot solve the optimal solution of the transformed problem without a QP solver. Intuitively speaking, if $N_{11}$ is not of full rank, we will drop some equality constraints, which might cause the solution to be relaxed, thus requiring us to solve the transformed QP problem with a QP solver.

Let us consider $N_{12}$. We rewrite the second equation of Eq. (8) as

$$\begin{bmatrix} N_{11} & N_{12} \\ N_{21} & N_{22} \end{bmatrix} \begin{bmatrix} A_a \\ A_{\bar{a}} \end{bmatrix} x^* = \begin{bmatrix} N_{11} & N_{12} \\ N_{21} & N_{22} \end{bmatrix} \begin{bmatrix} b_a \\ b_{\bar{a}} - s_{\bar{a}}^* \end{bmatrix}. \tag{13}$$

The first equation leads to

$$N_{11} A_a x^* + N_{12} A_{\bar{a}} x^* = N_{11} b_a + N_{12} b_{\bar{a}} - N_{12} s_{\bar{a}}^*, \tag{14}$$

which is equivalent to constructing new equality constraints with a linear combination of current constraints. We observe that $x^*$ remains for the transformed problem if and only if $N_{12} s_{\bar{a}}^* = 0$. A special case is $N_{12} = 0$, which removes the effect of inactive constraints on active ones. Otherwise, we must resolve this equation for $x^*$. Intuitively speaking, the margins of inactive constraints $s^*$ might relax or tighten an existing equation constraint, and we will have to solve the linear equation Eq. (13) rather than lazily copying the $x^*$ for the new problem.

The condition of $N_{22}$ is from the fact that if we multiply a negative constant on an inequality $a \leq b$, the inequality direction will flip. Let us look at the second equation of Eq. (13). We shall have

$$N_{21} A_a x^* + N_{22} A_{\bar{a}} x^* = N_{21} b_a + N_{22} (b_{\bar{a}} - s_{\bar{a}}^*), \tag{15}$$

given $A_a x^* = b_a$, we have

$$N_{22} A_{\bar{a}} x^* = N_{22} (b_{\bar{a}} - s_{\bar{a}}^*) \tag{16}$$

must hold. The matrix $N_{22}$ gives us combinations for a set of new inequality constraints given the existing constraints. Let us take out a row $v^\intercal$ from the $N_{22}$,

$$v^\intercal A_{\bar{a}} x^* = v^\intercal b_{\bar{a}} - v^\intercal s_{\bar{a}}^*, \tag{17}$$

yields a vector $a^\intercal = v^\intercal A_{\bar{a}}$ on the LHS and a scalar $b = v^\intercal b_{\bar{a}}$ on the RHS. We know that for a new inequality constraint, we should guarantee $a^\intercal x^* \leq b$, so $v^\intercal s_{\bar{a}}^* \geq 0$ must be satisfied. Generalizing this to all rows of $N_{22}$, we should have $N_{22} s_{\bar{a}}^* \geq 0$. $\qquad\square$

## D.2 Proof for Proposition 2.4

**Lemma D.1.** *KKT conditions are sufficient and necessary for LCQPs.*

See Boyd and Vandenberghe [2004, p. 244]. LCQPs satisfy the Slater conditions.

*Proof.* We extend the variable set by isolating the target variable $x'$ and rewriting the problem with variables $\begin{bmatrix} \boldsymbol{x} \\ x' \end{bmatrix} \in \mathbb{R}^{n+1}$, and the corresponding coefficients as $\boldsymbol{Q}' = \begin{bmatrix} \boldsymbol{Q} & \boldsymbol{q} \\ \boldsymbol{q}^\mathsf{T} & q \end{bmatrix} \in \mathbb{R}^{(n+1)\times(n+1)}$, $\boldsymbol{A}' = [\boldsymbol{A} \quad \boldsymbol{a}] \in \mathbb{R}^{m\times(n+1)}$, and $\boldsymbol{c}' = \begin{bmatrix} \boldsymbol{c} \\ c \end{bmatrix} \in \mathbb{R}^{n+1}, \boldsymbol{b} \in \mathbb{R}^m$.

**(If direction)** Suppose $x'^* = 0$, we can drop it as well as its corresponding coefficients $\boldsymbol{q}, q, \boldsymbol{a}, c$. We shall have the KKT conditions Eqs. (3a) and (3b) satisfied for the optimal solutions $\begin{bmatrix} \boldsymbol{x}^* \\ x'^* \end{bmatrix}$ and $\boldsymbol{\lambda}^*$,

$$\boldsymbol{Q}' \begin{bmatrix} \boldsymbol{x}^* \\ 0 \end{bmatrix} + \boldsymbol{A}'^\mathsf{T} \boldsymbol{\lambda}^* + \boldsymbol{c}' = \boldsymbol{0}$$

$$\boldsymbol{A}' \begin{bmatrix} \boldsymbol{x}^* \\ 0 \end{bmatrix} \leq \boldsymbol{b}.$$

We can introduce a row selection matrix $\boldsymbol{M}_1 := [\boldsymbol{I} \quad \boldsymbol{0}] \in \{0,1\}^{n\times(n+1)}$, which effectively drops the last row, and projects the KKT conditions as

$$\boldsymbol{M}_1 \boldsymbol{Q}' \boldsymbol{M}_1^\mathsf{T} \boldsymbol{M}_1 \begin{bmatrix} \boldsymbol{x}^* \\ 0 \end{bmatrix} + \boldsymbol{M}_1 \boldsymbol{A}'^\mathsf{T} \boldsymbol{\lambda}^* + \boldsymbol{M}_1 \boldsymbol{c}' = \boldsymbol{Q}\boldsymbol{x}^* + \boldsymbol{A}^\mathsf{T}\boldsymbol{\lambda}^* + \boldsymbol{c} = \boldsymbol{0}$$

$$\boldsymbol{A}' \boldsymbol{M}_1^\mathsf{T} \boldsymbol{M}_1 \begin{bmatrix} \boldsymbol{x}^* \\ 0 \end{bmatrix} = \boldsymbol{A}\boldsymbol{x}^* \leq \boldsymbol{b}.$$

Thus, removing $x'$ along with its associated coefficients preserves primal and dual feasibility, and $\boldsymbol{x}^*$ remains optimal for the reduced problem.

**(Only if direction)** Suppose we can remove $x'$ without affecting the optimality of $\boldsymbol{x}^*$. Then the reduced and full KKT conditions must be consistent for all possible coefficients associated with $x'$. That is,

$$\boldsymbol{Q}\boldsymbol{x}^* + \boldsymbol{A}^\mathsf{T}\boldsymbol{\lambda}^* + \boldsymbol{c} = \boldsymbol{0}$$
$$\boldsymbol{Q}\boldsymbol{x}^* + \boldsymbol{q}x'^* + \boldsymbol{A}^\mathsf{T}\boldsymbol{\lambda}^* + \boldsymbol{c} = \boldsymbol{0}$$
$$\boldsymbol{q}^\mathsf{T}\boldsymbol{x}^* + qx'^* + \boldsymbol{a}^\mathsf{T}\boldsymbol{\lambda}^* + c' = \boldsymbol{0}$$
$$\boldsymbol{A}\boldsymbol{x}^* \leq \boldsymbol{b}$$
$$\boldsymbol{A}\boldsymbol{x}^* + \boldsymbol{a}x'^* \leq \boldsymbol{b}.$$

For these conditions to hold for *all* choices of $\boldsymbol{q}, q, \boldsymbol{a}$, it must be that $x'^* = 0$; otherwise, the residual terms would depend on those coefficients and the solution would vary. Hence, $x'^* = 0$ is necessary for the removal to be valid without affecting the remaining solution. $\square$

## D.3 Proof for Proposition 2.5

*Proof.* We rewrite the problem by appending the target constraint as an additional row. The constraint matrix becomes $\boldsymbol{A}' = \begin{bmatrix} \boldsymbol{A} \\ \boldsymbol{a}'^\mathsf{T} \end{bmatrix} \in \mathbb{R}^{(m+1)\times n}$ and $\boldsymbol{b}' = \begin{bmatrix} \boldsymbol{b} \\ b' \end{bmatrix} \in \mathbb{R}^{(m+1)}$, and the corresponding dual variables are $\boldsymbol{\lambda}' = \begin{bmatrix} \boldsymbol{\lambda} \\ \lambda' \end{bmatrix} \in \mathbb{R}^{(m+1)}$.

**(If direction)** Suppose the additional constraint is strictly inactive, i.e., $\lambda'^* = 0$. We can introduce a row selection matrix $\boldsymbol{M}_2 := [\boldsymbol{I} \quad \boldsymbol{0}] \in \{0,1\}^{m\times(m+1)}$. The KKT conditions for the full system are,

$$\boldsymbol{Q}\boldsymbol{x}^* + \boldsymbol{A}'^\mathsf{T}\boldsymbol{\lambda}'^* + \boldsymbol{c} = \boldsymbol{Q}\boldsymbol{x}^* + \boldsymbol{A}^\mathsf{T}\boldsymbol{\lambda}^* + \boldsymbol{a}'\lambda'^* + \boldsymbol{c} = \boldsymbol{0}$$

$$\begin{bmatrix} \boldsymbol{A} \\ \boldsymbol{a}'^\mathsf{T} \end{bmatrix} \boldsymbol{x}^* \leq \begin{bmatrix} \boldsymbol{b} \\ b' \end{bmatrix}.$$

Since $\lambda'^* = 0$, we can write.

$$Qx^* + A'^\mathsf{T} M_2^\mathsf{T} M_2 \lambda'^* + c = Qx^* + A^\mathsf{T}\lambda^* + c = 0$$

$$M_2 A'x^* = Ax^* \leq b = M_2 b'.$$

This corresponds to the KKT conditions of the original problem with the constraint removed. Thus, removing the inactive constraint does not affect optimality.

**(Only if direction)** Suppose the constraint is removed and the optimal solution $x^*$, $\lambda^*$ remains valid for all possible coefficients $a'$, $b'$. Then both of the following must hold,

$$Qx^* + A'^\mathsf{T}\lambda'^* + c = Qx + A^\mathsf{T}\lambda^* + a'\lambda'^* + c = 0$$

$$Qx + A^\mathsf{T}\lambda^* + c = 0.$$

Subtracting the two equations yields $a'\lambda'^* = 0$. For this to hold for arbitrary $a'$, it must be that $\lambda'^* = 0$. Hence, the constraint must have been inactive. $\qquad\square$

### D.4   Proof for Proposition 2.6

*Proof.* **(If direction)** Let $x' = \begin{bmatrix} x \\ x' \end{bmatrix}$ with $x'^* = 0$, we can verify that

$$M_1 Q M_1^\mathsf{T} x'^* + M_1 A^\mathsf{T}\lambda^* + M_1 c = M_1 \left(Qx^* + A^\mathsf{T}\lambda^* + c\right) = 0$$

$$A M_1^\mathsf{T} x'^* = Ax^* \leq b.$$

Thus, $(x'^*, \lambda^*)$ satisfies the KKT conditions for the transformed problem.

**(Only if direction)** The equation

$$A M_1^\mathsf{T} x'^* = Ax^* + Aqx'^* \leq b$$

must hold for all choices of $q$, therefore $x'^* = 0$. $\qquad\square$

### D.5   Proof for Proposition C.1

*Proof.* **(If direction)** Suppose $\lambda'^* = 0$, and the transformed KKT conditions

$$Qx^* + \begin{bmatrix} A^\mathsf{T} & m \end{bmatrix} \begin{bmatrix} \lambda^* \\ \lambda'^* \end{bmatrix} + c = Qx^* + A^\mathsf{T}\lambda^* + c = 0$$

$$M_2 Ax^* = M_2 b$$

hold.

**(Only if direction)** The equations of pre- and post-transformation KKT must hold

$$Qx^* + A^\mathsf{T}\lambda^* + c = 0$$

$$Qx^* + A^\mathsf{T} M_2^\mathsf{T} \begin{bmatrix} \lambda^* \\ \lambda'^* \end{bmatrix} + c = 0 \qquad (18)$$

for all choices of $m$, therefore, $\lambda' = 0$. $\qquad\square$

### D.6   Proof for Section C

Proof for Appendix C.4:

*Proof.* **(If direction)** Given an LCQP instance $I := (Q, A, b, c)$ and its optimal primal and dual solution $x^*, \lambda^*$, new variable $x'^* = 0$, the following holds,

$$\begin{bmatrix} Q & 0 \\ 0^\mathsf{T} & q \end{bmatrix} \begin{bmatrix} x^* \\ x'^* \end{bmatrix} + \begin{bmatrix} A^\mathsf{T} \\ a^\mathsf{T} \end{bmatrix} \lambda^* = - \begin{bmatrix} c \\ c' \end{bmatrix}$$

$$\begin{bmatrix} A & a \end{bmatrix} \begin{bmatrix} x^* \\ x'^* \end{bmatrix} = b - s^*.$$

**(Only if direction)** Similar to the proof of Proposition 2.6. $\qquad\square$

Proof for Corollary C.2.1:

*Proof.* (**If direction**) Let $\boldsymbol{x}' = \begin{bmatrix} \boldsymbol{x} \\ x' \end{bmatrix}$ with $x'^* = 0$, we have

$$\begin{bmatrix} \boldsymbol{Q} & \boldsymbol{0} \\ \boldsymbol{0}^\mathsf{T} & q \end{bmatrix} \begin{bmatrix} \boldsymbol{x}^* \\ x'^* \end{bmatrix} + \begin{bmatrix} \boldsymbol{A}^\mathsf{T} & \boldsymbol{0} \\ \boldsymbol{a}^\mathsf{T} & 1 \end{bmatrix} \begin{bmatrix} \boldsymbol{\lambda}^* \\ \lambda'^* \end{bmatrix} + \begin{bmatrix} \boldsymbol{c} \\ c' \end{bmatrix} = \boldsymbol{0}$$
$$\begin{bmatrix} \boldsymbol{A} & \boldsymbol{a} \\ \boldsymbol{0}^\mathsf{T} & 1 \end{bmatrix} \begin{bmatrix} \boldsymbol{x}^* \\ x'^* \end{bmatrix} \leq \begin{bmatrix} \boldsymbol{b} \\ 0 \end{bmatrix},$$

which is satisfiable for some $\lambda'^*$.

(**Only if direction**) The inequality,

$$\begin{bmatrix} \boldsymbol{A} & \boldsymbol{a} \end{bmatrix} \begin{bmatrix} \boldsymbol{x}^* \\ x'^* \end{bmatrix} \leq \boldsymbol{b}$$

must hold for any choices of $\boldsymbol{a}$, therefore $x'^* = 0$. □

Proof for Proposition C.3:

*Proof.* Prove by construction. We can have $\boldsymbol{M} := \boldsymbol{I}$, and $\boldsymbol{B}_{11}, \boldsymbol{B}_2 := \boldsymbol{0}$, $\boldsymbol{B}_{21} = \boldsymbol{B}_{12}^\mathsf{T} := \begin{bmatrix} \boldsymbol{0} & \boldsymbol{a} \\ \boldsymbol{0}^\mathsf{T} & 1 \end{bmatrix}$,

$\boldsymbol{\beta} := \begin{bmatrix} \boldsymbol{0} \\ -c' \\ \boldsymbol{0} \\ 0 \end{bmatrix}.$ □

# E   Additional experiments

Here, we provide results for additional experiments.

## E.1   Computational overhead

To address **Q4**, we compare the per-epoch training time for regular training, data augmentation, and contrastive pretraining. Since data augmentation runs on CPU and model training on GPU, their absolute computation times are not directly comparable. Instead, we report the training time (excluding validation) with and without augmentation to evaluate practical overhead. Results in Table 6 show the mean and standard deviation over 100 epochs, using batch size 32 (128 for pretraining) and `num_workers` $= 4$.

Table 6: Per-epoch training time (in seconds) over 100 epochs. Data augmentation adds negligible overhead. Pretraining uses a larger batch size and remains efficient.

| Aug. | LP | | | QP | | |
| --- | --- | --- | --- | --- | --- | --- |
| | 10% | 100% | Pretrain | 10% | 100% | Pretrain |
| No | $1.005_{\pm 0.097}$ | $7.752_{\pm 0.143}$ | N/A | $1.244_{\pm 0.076}$ | $10.374_{\pm 0.158}$ | N/A |
| Yes | $0.975_{\pm 0.061}$ | $7.822_{\pm 0.197}$ | $7.699_{\pm 0.151}$ | $1.235_{\pm 0.069}$ | $10.247_{\pm 0.343}$ | $9.648_{\pm 0.139}$ |

## E.2   Size generalization

For both LPs and QPs, we generate larger instances with increased numbers of variables and constraints (proportional to the original sizes) while maintaining a fixed average graph degree. The exact parameters are detailed in Appendix F. As shown in Table 7, increasing instance size improves performance across all methods, suggesting some extent of size generalization. Importantly, models pre-trained consistently outperform models trained from scratch across all problem sizes and levels of data scarcity.

Table 7: Generalization performance of contrastive pretrained MPNNs on larger LP and QP instances. Models pretrained with contrastive learning consistently outperform those trained from scratch.

| Size | Pretrained | LP 10% | LP 20% | LP 100% | QP 10% | QP 20% | QP 100% |
|---|---|---|---|---|---|---|---|
| 100% | No | $8.539_{\pm0.206}$ | $6.149_{\pm0.153}$ | $2.784_{\pm0.081}$ | $5.304_{\pm0.229}$ | $3.567_{\pm0.034}$ | $1.240_{\pm0.088}$ |
| | Yes | $3.472_{\pm0.086}$ | $2.794_{\pm0.049}$ | $1.588_{\pm0.056}$ | $3.791_{\pm0.097}$ | $2.427_{\pm0.083}$ | $0.926_{\pm0.031}$ |
| 150% | No | $6.707_{\pm0.052}$ | $4.968_{\pm0.095}$ | $2.431_{\pm0.032}$ | $4.253_{\pm0.166}$ | $2.947_{\pm0.070}$ | $0.986_{\pm0.002}$ |
| | Yes | $2.948_{\pm0.225}$ | $2.304_{\pm0.029}$ | $1.331_{\pm0.044}$ | $2.999_{\pm0.120}$ | $2.021_{\pm0.051}$ | $0.766_{\pm0.031}$ |
| 200% | No | $5.820_{\pm0.069}$ | $4.173_{\pm0.085}$ | $2.052_{\pm0.087}$ | $3.971_{\pm0.028}$ | $2.474_{\pm0.049}$ | $0.847_{\pm0.044}$ |
| | Yes | $2.624_{\pm0.091}$ | $2.122_{\pm0.102}$ | $1.188_{\pm0.048}$ | $2.688_{\pm0.103}$ | $1.793_{\pm0.103}$ | $0.654_{\pm0.033}$ |
| 250% | No | $5.275_{\pm0.082}$ | $3.759_{\pm0.049}$ | $1.912_{\pm0.023}$ | $3.708_{\pm0.089}$ | $2.382_{\pm0.065}$ | $0.775_{\pm0.030}$ |
| | Yes | $2.451_{\pm0.108}$ | $1.912_{\pm0.063}$ | $1.121_{\pm0.029}$ | $2.601_{\pm0.232}$ | $1.671_{\pm0.054}$ | $0.605_{\pm0.023}$ |

## E.3 Other pretraining

Besides contrastive pretraining, we also pretrain with supervised learning with and without augmentation on the random LP dataset, and test its OOD performance on the set cover dataset. As shown in Table 8, supervised pretraining performs surprisingly well, especially with data augmentation.

Table 8: Generalization performance of supervised pretrained MPNNs on OOD set cover instances.

| Pretrained | Ratio 10% | Ratio 20% | Ratio 100% |
|---|---|---|---|
| No | $5.297_{\pm0.787}$ | $2.531_{\pm0.861}$ | $0.752_{\pm0.042}$ |
| CL | $1.965_{\pm0.206}$ | $1.349_{\pm0.161}$ | $0.662_{\pm0.055}$ |
| Su. | $1.994_{\pm0.115}$ | $1.521_{\pm0.085}$ | $0.778_{\pm0.006}$ |
| Su.+aug. | $\mathbf{1.034_{\pm0.055}}$ | $\mathbf{0.848_{\pm0.045}}$ | $\mathbf{0.545_{\pm0.026}}$ |

## E.4 Ablation on temperature parameter

We conduct experiments on the temperature parameter $\tau$ in the contrastive loss $Eq.$ (11). We use the same setting as in Table 2, pick $\tau \in \{0.01, 0.1, 1\}$, and run it with our method on LP instances. The results are summarized as in Table 9. As shown, $\tau = 0.1$ is consistently the best option.

Table 9: Ablation on the temperature parameter $\tau$.

| $\tau$ | Ratio 10% | Ratio 20% | Ratio 100% |
|---|---|---|---|
| 0.01 | $3.483_{\pm0.074}$ | $2.903_{\pm0.078}$ | $1.671_{\pm0.053}$ |
| 0.1 | $\mathbf{3.472_{\pm0.086}}$ | $\mathbf{2.794_{\pm0.049}}$ | $\mathbf{1.588_{\pm0.056}}$ |
| 1 | $4.269_{\pm0.197}$ | $2.802_{\pm0.032}$ | $1.628_{\pm0.046}$ |

## E.5 QPLIB

We evaluate QPLIB [Furini et al., 2019]. Due to the limited size, large scale, and extreme heterogeneity of QPLIB instances, performing a standard train/validation/test split is infeasible. Existing works that use QPLIB for evaluation typically rely on perturbing problem coefficients [Wu et al., 2024, Yang et al., 2024b]. These approaches, however, have significant limitations:

1. The augmentations are not label-preserving and thus require solving each perturbed instance from scratch;

2. Perturbations may break feasibility, which limits the perturbation strength;

3. As a result, weaker perturbations are often used, but they yield training instances that are nearly identical to the test instances, making evaluation less meaningful.

Our proposed method, on the other hand, includes both structural and feature perturbation and is targeted at better generalization performance. For the current perturbation method and ours to work well, one can reduce the perturbation rate to an arbitrarily small value to fit the test data perfectly, which appears nonsensical. To study transferability and fitting efficiency on QPLIB, we conduct a pre-training experiment.

1. We generate a foundation dataset of 10000 large-scale random QP instances with sizes ranging from 1000 to 1500 variables and constraints, which are 100–200 times larger than the problems used in Table 2. Since no labels are needed for contrastive pretraining, data generation is efficient. Training takes around 30 seconds per epoch using 4 NVIDIA L40S GPUs.

2. We select feasible LCQP instances from QPLIB, relax integer constraints, and train an MPNN to fit these problems in a supervised setting. We compare models trained from scratch with models initialized from the pretrained weights.

Here are the problem statistics and results:

Table 10: Statistics of selected QPLIB instances.

| Name | cons. | vars. | A density | Q density |
|------|-------|-------|-----------|-----------|
| QPLIB_3694 | 3280 | 3240 | 0.001208 | 0.000313 |
| QPLIB_3708 | 12917 | 12930 | 0.000171 | 0.000628 |
| QPLIB_3861 | 4650 | 4530 | 0.000856 | 0.000222 |
| QPLIB_3871 | 1040 | 1025 | 0.003772 | 0.001000 |
| QPLIB_8559 | 5000 | 10000 | 0.000500 | 0.000700 |

We train each model for 100 epochs and compare predicted objectives to the actual optimal value. Longer training will diminish the advantage of pretraining, since we are testing on the same data we used for training. We repeat the training 3 times with random seeds and report the mean.

Table 11: Training performance on selected QPLIB instances. Pretraining substantially improves convergence.

| Name | Pretrain | Obj.(optimal) | Obj.(predict) |
|------|----------|---------------|---------------|
| QPLIB_3694 | No | 0.000 | 10.362 |
|            | Yes | 0.000 | 0.142 |
| QPLIB_3708 | No | -42.469 | -42.469 |
|            | Yes | -42.469 | -42.469 |
| QPLIB_3861 | No | 0.000 | 10.895 |
|            | Yes | 0.000 | -0.193 |
| QPLIB_3871 | No | 0.000 | 7.205 |
|            | Yes | 0.000 | -0.193 |
| QPLIB_8559 | No | 15.793 | 29.208 |
|            | Yes | 15.793 | 12.331 |

As shown in Table 11, pretrained models generally fit faster and more accurately within limited training epochs. QPLIB_3708 is an exception, though both models converge very close to the actual objective. These results support the scalability and transferability of our method.

# F   Hyperparameters

This section lists some crucial hyperparameters except those already described in Section 3.

In supervised learning, we generate augmented data on the fly and control the degree of perturbation using an *augmentation strength* parameter $\alpha$. For example, a node-dropping strength of $\alpha = 0.3$ removes 30% of the graph's nodes. We treat the variable-constraint graph as homogeneous for baselines from You et al. [2020]. For scaling-based augmentations (e.g., variable or constraint coefficients), we interpret strength $\alpha$ as multiplication by $e^{\alpha}$. For constraint addition, we add a fraction $\alpha$ of new

constraints, each a convex combination of three existing constraints. To balance original and augmented views, we sample a scaled strength $\alpha' := \alpha\epsilon$, where $\epsilon \sim \mathcal{U}(0, 1)$, ensuring smooth interpolation between unaltered and fully augmented data. The strengths used in our experiments are listed below.

Table 12: Augmentation strength of various augmentations in Table 1.

| Augmentation | LP | QP |
|---|---|---|
| Drop node | 0.10 | 0.05 |
| Mask node | 0.10 | 0.05 |
| Flip edge | 0.10 | 0.05 |
| Drop vars. | 0.99 | 0.99 |
| Drop cons. | 0.99 | 0.99 |
| Scale cons. | 1.00 | 1.00 |
| Scale vars. | 1.00 | 1.00 |
| Add cons. | 0.50 | 0.50 |
| Add vars. | 0.80 | 0.60 |

For the combined augmentations, we use the following combination for both LP and QP, and for each instance, we sample two augmentations from the list below.

Table 13: Augmentation strength of combined augmentations in Table 1.

| Augmentation | Strength |
|---|---|
| Drop vars. | 0.00 |
| Drop cons. | 0.50 |
| Scale cons. | 0.50 |
| Scale vars. | 0.50 |
| Add cons. | 0.60 |
| Add vars. | 0.00 |

For contrastive pretraining, we omit interpolation and apply fixed augmentation strengths. For GraphCL [You et al., 2020], we tune the strengths of its three components: for LP, we use 0.2 node dropping, 0.1 edge flipping, and 0.13 feature masking; for QP, 0.3, 0.17, and 0.15, respectively. For GCC [Qiu et al., 2020], we use a random walk length of 50 for LP and 200 for QP. IGSD [Zhang et al., 2023] and MVGRL [Hassani and Khasahmadi, 2020] apply graph diffusion with 10% edge addition. InfoGraph [Sun et al., 2019], DGI [Veličković et al., 2018], and GAE [Kipf and Welling, 2016] require no additional hyperparameters. For our method, we use the following configuration.

Table 14: Augmentation strength of combined augmentations in Table 2.

| Augmentation | LP | QP |
|---|---|---|
| Drop cons. (heuristic) | 0.05 | 0.07 |
| Scale cons. | 0.40 | 1.03 |
| Scale vars. | 1.07 | 0.65 |
| Add cons. | 0.36 | 0.33 |
| Add vars. | 0.46 | 0.26 |

# G   Data generation

We use the following pseudo code for random LP and QP generation; see Algorithms 1 and 2.

For the instances used in Tables 1 and 2 and Table 7, we generate the instances with sizes as described in Table 15.

For the OOD LP instances in Table 3, we follow the generation procedure in Gasse et al. [2019]. The idea is to keep the dimensions and the density of the matrix $\boldsymbol{A}$ similar to those of the problems we pretrain on. Hence, we generate with 100 rows and columns and 5% $\boldsymbol{A}$ matrix density for SC; 100

---

**Algorithm 1** Generate a sparse feasible LP instance.

---

**Require:** Number of constraints $m$, number of variables $n$, constraint density $\rho_A$, random seed
**Ensure:** Matrices $(c, A, b)$ defining an LP
1: Initialize random number generator `rng` with the given seed
2: Generate sparse random matrix $A \sim \mathcal{N}(0,1)^{m \times n}$ with density $\rho_A$
3: Sample $x_{\text{raw}} \sim \mathcal{N}(0,1)^n$
4: Set $x := |x_{\text{raw}}|$ (elementwise absolute value)
5: Compute $s_{\text{noise}} \sim \mathcal{N}(0,1)^m$ (slack noise)
6: Set $b := Ax + |s_{\text{noise}}|$
7: Sample random linear term $c \sim \mathcal{N}(0,1)^n$
8: **return** $(c, A, b)$

---

---

**Algorithm 2** Generate a sparse feasible QP instance.

---

**Require:** Number of constraints $m$, number of variables $n$, constraint density $\rho_A$, matrix density $\rho_P$, random seed
**Ensure:** Matrices $(Q, c, A, b)$ defining a QP
1: Initialize random number generator `rng` with the given seed
2: Generate sparse random matrix $A \sim \mathcal{N}(0,1)^{m \times n}$ with density $\rho_A$
3: Sample $x_{\text{raw}} \sim \mathcal{N}(0,1)^n$
4: Set $x := |x_{\text{raw}}|$ (elementwise absolute value)
5: Compute $s_{\text{noise}} \sim \mathcal{N}(0,1)^m$ (slack noise)
6: Set $b := Ax + |s_{\text{noise}}|$
7: Sample random linear term $c \sim \mathcal{N}(0,1)^n$
8: Generate sparse positive semi-definite matrix $Q \in \mathbb{R}^{n \times n}$ using SciPy `make_sparse_spd_matrix` with sparsity parameter $\alpha = 1 - \rho_P$
9: **return** $(Q, c, A, b)$

---

nodes and 0.02 edge probability for MIS; 100 items and 100 bids for CA; and 25 customers, three facilities, 0.5 ratio for CFL problem.

For the OOD QP instances, we generate SVM problems with Algorithm 3 with 100 samples and 0.05 density.

---

**Algorithm 3** Generate soft-margin SVM QP instance.

---

**Require:** Number of samples $n$, number of features $d$, regularization parameter $\lambda$, feature density $\rho$, random number generator `rng`
**Ensure:** Matrices $(Q, c, A, b)$ defining a QP
1: Generate positive samples matrix $A_1 \sim \mathcal{N}\left(\frac{1}{d\rho}, \frac{1}{d\rho}\right)^{\frac{n}{2} \times d}$
2: Generate negative samples matrix $A_2 \sim \mathcal{N}\left(-\frac{1}{d\rho}, \frac{1}{d\rho}\right)^{\frac{n}{2} \times d}$
3: Stack data: $A := \begin{bmatrix} A_1 \\ A_2 \end{bmatrix}$
4: Sparsify $A$ with density $\rho$
5: Construct label vector $y := [1, \ldots, 1, -1, \ldots, -1] \in \{-1, 1\}^n$
6: Apply labels to data: $A := A \cdot y^{\mathsf{T}}$
7: Form constraint matrix: $A := -[A \ \ I_n]$
8: Set right-hand side: $b := -\mathbf{1}_n$
9: Define quadratic matrix $Q := \text{diag}([1, \ldots, 1, 0, \ldots, 0]) \in \{0, 1\}^{(d+n) \times (d+n)}$
10: Define linear term: $c := [0, \ldots, 0, \lambda, \ldots, \lambda] \in \{0, \lambda\}^{d+n}$
11: **return** $(Q, c, A, b)$

---

We generate portfolio problems as Algorithm 4, with 100 assets and 0.05 density.

We generate LASSO problems using Algorithm 5 with 50 samples, 50 features, and a density of 0.05.

Table 15: Random instance generation hyperparameters.

| Size | | LP | | | | QP | | |
|------|------|------|-------------|------|------|-------------|-------------|
| | #Col | #Row | $A$ density | #Col | #Row | $A$ density | $Q$ density |
| 100% | 100 | 100 | 0.05 | 100 | 100 | 0.05 | 0.05 |
| 150% | 150 | 150 | 0.03 | 150 | 150 | 0.03 | 0.03 |
| 200% | 200 | 200 | 0.025 | 200 | 200 | 0.025 | 0.025 |
| 250% | 250 | 250 | 0.02 | 250 | 250 | 0.02 | 0.02 |

---

**Algorithm 4** Generate mean-variance portfolio QP instance.

---

**Require:** Number of assets $n$, covariance matrix density $\rho$, random number generator `rng`
**Ensure:** Matrices $(Q, c, A, b, A_{\text{eq}}, b_{\text{eq}})$ defining a QP
1: Generate sparse positive definite covariance matrix $Q \in \mathbb{R}^{n \times n}$ using `make_sparse_spd_matrix` with $\alpha = 1 - \rho$, and eigenvalues in $[0.1, 0.9]$
2: Set linear cost vector $c := \mathbf{0}_n$
3: Generate inequality constraint matrix $A \sim \mathcal{N}(0, 0.01)^{1 \times n}$
4: Generate equality constraint matrix $A_{\text{eq}} := 0.01 \cdot \mathbf{1}_n^\top$
5: Set inequality right hand side: $b := [-1]$
6: Set equality right hand side: $b_{\text{eq}} := [1]$
7: **return** $(Q, c, A, b, A_{\text{eq}}, b_{\text{eq}})$

---

---

**Algorithm 5** Generate LASSO QP instance.

---

**Require:** Number of samples $n$, number of features $d$, feature density $\rho$, regularization parameter $\lambda$, random number generator `rng`
**Ensure:** Matrices $(Q, c, A, b)$ defining a QP
1: Generate sparse design matrix $X \sim \mathcal{N}(0, 1)^{n \times d}$ with density $\rho$
2: Sample true weight vector $w_{\text{true}} \sim \mathcal{N}(0, 1)^d$
3: Generate noise vector $\epsilon \sim \mathcal{N}(0, 0.5)^n$
4: Compute target: $y := X w_{\text{true}} + \epsilon$
5: Compute quadratic term: $Q_0 := \frac{1}{2} X^\top X$
6: Compute linear term: $c_0 := -X^\top y$
7: Form block matrix: $Q := \begin{bmatrix} Q_0 & \mathbf{0} \\ \mathbf{0} & \mathbf{0} \end{bmatrix} \in \mathbb{R}^{2d \times 2d}$
8: Form cost vector: $c := \begin{bmatrix} c_0 \\ \lambda \cdot \mathbf{1}_d \end{bmatrix}$
9: Construct constraint matrix:
$$A := \begin{bmatrix} -I_d & -I_d \\ I_d & -I_d \end{bmatrix}$$
10: Set constraint right-hand side: $b := \mathbf{0}_{2d}$
11: **return** $(Q, c, A, b)$

---

