# OpenReview forum: "Principled Data Augmentation for Learning to Solve Quadratic Programming Problems"
_NeurIPS.cc/2025/Conference — NeurIPS 2025 spotlight_

### Official Review · Reviewer_ahYQ · 2025-06-21

**Clarity:** 4
**Significance:** 4
**Originality:** 4
**Rating:** 6
**Confidence:** 3

**Summary:**

This paper presents a solid and comprehensive analysis of data augmentation for both linear programming (LP) and quadratic programming (QP) problems. The authors propose a rigorous theoretical framework for identifying valid data augmentations, grounded in dual problem formulations. Importantly, the method maintains computational efficiency, making it practical for large-scale applications. Leveraging these theoretical guarantees, the authors generate additional data to enhance both supervised learning and pretraining tasks, effectively addressing the issue of limited original data. Experimental results are compelling, demonstrating substantial performance improvements with comparable computational costs to training without data augmentation.

**Questions:**

See Weaknesses.

**Ethical Concerns:**

["NO or VERY MINOR ethics concerns only"]

**Final Justification:**

I think the work has valuable impact so I raise my score.

**Limitations:**

None.

**Paper Formatting Concerns:**

None.

**Quality:**

4

**Strengths And Weaknesses:**

Strengths:

1. The theoretical foundation is solid, and the proposed data augmentation method effectively addresses the primal problem under limited data settings.

2. The experimental results are strong, demonstrating the practical effectiveness of the approach across various benchmarks.

Weaknesses:

1. Can the author provide some experimental results on public datasets such as QPLIB?

2. The proposed data augmentation guarantees that the solution remains unchanged. I am curious whether there exists a way to design augmentations that introduce slight variations in the solution, in order to obtain more diverse augmented data?

---

> ### Author Rebuttal · Authors · 2025-07-30
>
> We sincerely thank the reviewer for the thoughtful and encouraging review. We are grateful for your recognition of both the theoretical contributions and the practical effectiveness of our method. Below, we address your questions in detail.
> ### **Experiments on public datasets such as QPLIB**
> Thank you for this suggestion. We fully acknowledge this limitation. We do an evaluation on QPLIB. Due to the limited size, large scale, and extreme heterogeneity of QPLIB instances, performing a standard train/validation/test split is infeasible. Existing works that use QPLIB for evaluation typically rely on perturbing problem coefficients [1][2]. These approaches, however, have significant limitations:
> 1. The augmentations are not label-preserving and thus require solving each perturbed instance from scratch;
> 2. Perturbations may break feasibility, which limits the perturbation strength;
> 3. As a result, weaker perturbations are often used, but they yield training instances that are nearly identical to the test instances, making evaluation less meaningful.
>
> Our proposed method, on the other hand, includes both structural and featural perturbation and is targeted at better generalization performance. For the current perturbation method and ours to work well, one can reduce the perturbation rate to an arbitrarily small value to fit perfectly on the test data, but that isn't very meaningful. To study transferability and fitting efficiency on QPLIB, we conduct a pretraining experiment:
> -  Step 1: We generate a foundation dataset of 10000 large-scale random QP instances with sizes ranging from 1000 to 1500 variables and constraints, being 100–200 times larger than the problems used in Table 2. Since no labels are needed for contrastive pretraining, data generation is efficient. Training takes around 30 seconds per epoch using 4 NVIDIA L40S GPUs.
> -  Step 2: We select feasible LCQP instances from QPLIB that vary a lot in size, relax integer constraints, and train an MPNN to fit these problems in a supervised setting. We compare models trained from scratch with models initialized from the pretrained weights.
>
> Here are the problem statistics and results:
>
> | Name | cons. | vars. | A density | Q density |
> |-------------|--------|-------|-----------|-----------|
> | QPLIB_3694 | 3280 | 3240 | 0.001208 | 0.000313 |
> | QPLIB_3708 | 12917 | 12930 | 0.000171 | 0.000628 |
> | QPLIB_3861 | 4650 | 4530 | 0.000856 | 0.000222 |
> | QPLIB_3871 | 1040 | 1025 | 0.003772 | 0.001000 |
> | QPLIB_8559 | 5000 | 10000 | 0.000500 | 0.000700 |
>
> We train each model for 100 epochs and compare predicted objectives to the actual optimal value. Longer training will diminish the advantage of pretraining, as we are testing exactly on the training data. We repeat the pretraining and fine-tuning with 3 random seeds and report the mean.
>
> | ID | Pretrain | Obj.(optimal) | Obj.(predict) @ 100th epoch |
> |-------------|----------|---------------|---------------|
> | QPLIB_3694 | No | 0.0 | 10.36161 |
> | QPLIB_3694 | Yes | 0.0 | __0.14176__ |
> | QPLIB_3708 | No | -42.46981 | __-42.46981__ |
> | QPLIB_3708 | Yes | -42.46981 | -42.46916 |
> | QPLIB_3861 | No | 0.0 | 10.89544 |
> | QPLIB_3861 | Yes | 0.0 | __0.41946__ |
> | QPLIB_3871 | No | 0.0 | 7.20569 |
> | QPLIB_3871 | Yes | 0.0 | __-0.19335__ |
> | QPLIB_8559 | No | 15.79288 | 29.20809 |
> | QPLIB_8559 | Yes | 15.79288 | __12.33148__ |
>
> As shown, pretrained models generally fit faster and more accurately within limited training epochs. QPLIB_3708 is an exception, though both models converge very close to the true objective. These results support the scalability and transferability of our method.
>
> [1] Wu, Chenyang, et al. "On representing convex quadratically constrained quadratic programs via graph neural networks." arXiv preprint arXiv:2411.13805 (2024).
> [2] Yang, Linxin, et al. "An Efficient Unsupervised Framework for Convex Quadratic Programs via Deep Unrolling." arXiv preprint arXiv:2412.01051 (2024).
>
> ### **Regarding augmentations that introduce slight variations**
> This is an excellent and insightful idea. Indeed, Equation (7) in the main paper defines a highly expressive framework that, in principle, covers the whole space of linearly-constrained quadratic programs (LCQP), including those that induce slight deviations in the solution. However, incorporating such deviations into data augmentation presents several practical and theoretical challenges:
> - **In the supervised setting**, even a small change in the solution still requires solving the new problem, making it as computationally expensive as generating an entirely new instance. Thus, the advantage of analytic recoverability of our framework is lost.
> - **In the self-supervised setting**, introducing slight solution changes raises the question of **how much** deviation is acceptable. In other words, it is unclear when an augmented instance retains the same semantic meaning. This introduces additional hyperparameter tuning and problem-specific calibration. While minor deviations may be harmless in practice, the boundary is unclear and complicated to formalize.
> We do explore this direction heuristically in Appendix D.3, where we describe an augmentation strategy that may slightly alter the solution while maintaining approximate correctness. This heuristic is included in our contrastive pretraining experiments (Table 2), and no harm in empirical performance is observed.
> Generally, perturbing continuous optimization problems is fundamentally different from perturbing problems with discrete labels. In our case, even a small change to the problem data can yield a quantitatively different solution, and solvers are generally not designed to exploit similarity between near-identical instances unless warm-starting is explicitly supported.
> We believe your suggestion opens an interesting topic for future work. Developing solution-perturbing augmentations with provable guarantees could greatly enrich the current augmentation space, especially for self-supervised learning.
>
> **We appreciate the reviewer’s thoughtful comments and valuable suggestions. We hope our detailed responses have sufficiently addressed the concerns and helped clarify the contributions and scope of our work.**

---

### Official Review · Reviewer_PXtd · 2025-06-23

**Clarity:** 2
**Significance:** 2
**Originality:** 2
**Rating:** 4
**Confidence:** 5

**Summary:**

This paper proposes a theoretically motivated data augmentation framework for LP/QP tasks by transforming the KKT system, enabling supervised and contrastive training.

**Questions:**

1. Please provide OOD evaluation in the supervised setting under data augmentation to isolate the benefit of augmentation itself, beyond contrastive pretraining.

2. Can the authors release code and data generation scripts to ensure full reproducibility?

3. How does performance vary with the number of augmented instances used during training? I suggest reporting results under different augmentation intensities.

4. How does this work compare empirically and conceptually with MILP-FBGen and similar LP instance generation work?

**Ethical Concerns:**

["NO or VERY MINOR ethics concerns only"]

**Final Justification:**

The rebuttal addresses my concern. So i raise the score.

**Limitations:**

See above

**Quality:**

2

**Strengths And Weaknesses:**

Strengths:

1. Clear theoretical foundation via affine KKT transformations.

2. Augmentations are label-preserving or solution-independent, enabling efficient integration into both learning paradigms.

3. Extensive experiments show strong gains under data scarcity and OOD transfer.

Weaknesses:

1. Experiments are limited to synthetic, fixed-scale problems; scalability remains unclear.

2. The number of augmented instances during training is not analyzed, making it hard to assess the effect of augmentation intensity.

3. In the LP setting, similar augmentation motivations have been explored (e.g., MILP-FBGen [1]), limiting novelty.

[1] MILP-FBGen: LP/MILP Instance Generation with Feasibility/Boundedness

---

> ### Author Rebuttal · Authors · 2025-07-30
>
> We thank the reviewer for the thoughtful and insightful suggestions, which we address below.
>
> ### **Regarding synthetic datasets**
> We fully acknowledge this limitation. We do an evaluation on QPLIB. Due to the limited size, large scale, and extreme heterogeneity of QPLIB instances, performing a standard train/validation/test split is infeasible. Existing works that use QPLIB for evaluation typically rely on perturbing problem coefficients [1][2]. These approaches, however, have significant limitations:
> 1. The augmentations are not label-preserving and thus require solving each perturbed instance from scratch;
> 2. Perturbations may break feasibility, which limits the perturbation strength;
> 3. As a result, weaker perturbations are often used, but they yield training instances that are nearly identical to the test instances, making evaluation less meaningful.
>
> Our proposed method, on the other hand, includes both structural and featural perturbation and is targeted at better generalization performance. For the current perturbation method and ours to work well, one can reduce the perturbation rate to an arbitrarily small value to fit perfectly on the test data, but that isn't very meaningful. To study transferability and fitting efficiency on QPLIB, we conduct a pretraining experiment:
> -  Step 1: We generate a foundation dataset of 10000 large-scale random QP instances with sizes ranging from 1000 to 1500 variables and constraints, being 100–200 times larger than the problems used in Table 2. Since no labels are needed for contrastive pretraining, data generation is efficient. Training takes around 30 seconds per epoch using 4 NVIDIA L40S GPUs.
> -  Step 2: We select feasible LCQP instances from QPLIB that vary a lot in size, relax integer constraints, and train an MPNN to fit these problems in a supervised setting. We compare models trained from scratch with models initialized from the pretrained weights.
>
> Here are the problem statistics and results:
>
> | Name | cons. | vars. | A density | Q density |
> |-------------|--------|-------|-----------|-----------|
> | QPLIB_3694 | 3280 | 3240 | 0.001208 | 0.000313 |
> | QPLIB_3708 | 12917 | 12930 | 0.000171 | 0.000628 |
> | QPLIB_3861 | 4650 | 4530 | 0.000856 | 0.000222 |
> | QPLIB_3871 | 1040 | 1025 | 0.003772 | 0.001000 |
> | QPLIB_8559 | 5000 | 10000 | 0.000500 | 0.000700 |
>
> We train each model for 100 epochs and compare predicted objectives to the actual optimal value. Longer training will diminish the advantage of pretraining, as we are testing exactly on the training data. We repeat the pretraining and fine-tuning with 3 random seeds and report the mean.
>
> | ID | Pretrain | Obj.(optimal) | Obj.(predict) @ 100th epoch |
> |-------------|----------|---------------|---------------|
> | QPLIB_3694 | No | 0.0 | 10.36161 |
> | QPLIB_3694 | Yes | 0.0 | __0.14176__ |
> | QPLIB_3708 | No | -42.46981 | __-42.46981__ |
> | QPLIB_3708 | Yes | -42.46981 | -42.46916 |
> | QPLIB_3861 | No | 0.0 | 10.89544 |
> | QPLIB_3861 | Yes | 0.0 | __0.41946__ |
> | QPLIB_3871 | No | 0.0 | 7.20569 |
> | QPLIB_3871 | Yes | 0.0 | __-0.19335__ |
> | QPLIB_8559 | No | 15.79288 | 29.20809 |
> | QPLIB_8559 | Yes | 15.79288 | __12.33148__ |
>
> As shown, pretrained models generally fit faster and more accurately within limited training epochs. QPLIB_3708 is an exception, though both models converge very close to the actual objective. These results support the scalability and transferability of our method.
>
> [1] Wu, Chenyang, et al. "On representing convex quadratically constrained quadratic programs via graph neural networks." arXiv preprint arXiv:2411.13805 (2024).
> [2] Yang, Linxin, et al. "An Efficient Unsupervised Framework for Convex Quadratic Programs via Deep Unrolling." arXiv preprint arXiv:2412.01051 (2024).
>
> ### **Regarding scalability**
> Scalability is well addressed in our framework. We introduced the notion of _efficiently recoverable transformations_, whose concrete instantiations have provably linear time complexity. Empirically, we demonstrate this through our large-scale pretraining experiment on 10000 QP instances with 1000–1500 variables and constraints, as shown above for QPLIB. Since no solver is required during pretraining, the dataset creation and runtime are efficient.
>
> ### **Augmentation intensity**
> This is an excellent question. Without loss of generality, we generate one augmented instance per data point per epoch, as required by contrastive learning. For the supervised setting, we conducted new experiments, using two of the best-performing augmentations on LPs: dropping inactive constraints and scaling variable coefficients. Using the same setup as Table 1, we pick the number of augmentations per instance in {1, 2, 3}.
>
> The results are quite counterintuitive. Interestingly, more augmentations do **not** help, except slightly in the 100% training data case. In fact, they slightly hurt performance and slow down the convergence, as we have observed in the training curves (unfortunately, we cannot provide external links or plots). From both effectiveness and efficiency standpoints, using a single augmentation per instance seems optimal.
>
> - Dropping inactive constraints
>
> | #Samples | LP 10% | LP 20% | LP 100% |
> |----------|---------------|---------------|---------------|
> | 1 |__5.563±0.097__|__3.691±0.087__| 1.656±0.057 |
> | 2 | 6.135±0.267 | 4.023±0.055 |__1.596±0.054__|
> | 3 | 6.277±0.136 | 4.092±0.029 | 1.754±0.062 |
>
> - Scaling variables
>
> | #Samples | LP 10% | LP 20% | LP 100% |
> |----------|---------------|-----------------|-----------------|
> | 1 |__4.682±0.162__| __3.208±0.116__ | 1.241±0.052 |
> | 2 | 5.027±0.113 | 3.426±0.097 | __1.204±0.033__ |
> | 3 | 5.068±0.130 | 3.518±0.103 | 1.284±0.060 |
>
> ### **OOD evaluation in the supervised setting**
> This is a good suggestion, and we will include the corresponding results in the revised paper. We pretrain on random LP instances with and without augmentation, then transfer the model to OOD LP instances—specifically, the set cover problem. For comparison, we also include results from the original paper: training from scratch on OOD data, and using contrastive pretraining.
>
> The results show that both supervised pretraining (with/without augmentation) and contrastive pretraining outperform training from scratch. Remarkably, supervised pretraining with augmentation yields the best performance, highlighting the benefit of our proposed augmentations. Nonetheless, the label-free nature of contrastive pretraining still offers practical advantages.
>
> | Pretrain | SC 10% | SC 20% | SC 100% |
> |--------------|---------------|---------------|-----------------|
> | None | 5.297±0.787 | 2.531±0.861 | 0.752±0.042 |
> | CL | 1.965±0.206 | 1.349±0.161 | 0.662±0.055 |
> | Supervised | 1.994±0.115 | 1.521±0.085 | 0.778±0.006 |
> |Supervised+aug|__1.034±0.055__|__0.848±0.045__| __0.545±0.026__ |
>
> ### **Comparing to other MILP generation work**
> We thank the reviewer for pointing out these related works. This is a sharp and knowledgeable observation. Indeed, recent MILP instance generation methods such as MILP-FBGen address data scarcity in learning-based optimization, but their objectives and methodologies are fundamentally different from ours. We have already conducted a literature survey for these methods in Appendix C.
>
> While both our work and the related work aim to enrich training data, their approaches focus on generating new instances that are distributionally similar to the original problem set, typically using generative models or with strong assumptions such as diagonalizable matrices. These methods often ignore labels and may not guarantee feasibility or boundedness without strong structural assumptions. Moreover, solving these generated MILPs still requires a solver, adding to the computational cost. These limitations make it hard to compare our method to theirs empirically.
>
> In contrast, our approach is mathematically grounded, learning-free, and solver-free. We design label-preserving augmentations by transforming the KKT system of a given LP/QP instance, enabling us to generate semantically equivalent problems—i.e., those with the same solution and objective. This is crucial for both supervised learning, where we require efficient label computation, and self-supervised contrastive pretraining, where we need two solution-preserving views of the same instance. These transformations are derived analytically and allow us to recover solutions efficiently, without retraining a model or solving the problem again. Besides, we do not focus on preserving the structural distribution of the generated instances, which can potentially cover a broader spectrum of instances.
>
> ### **Code release and data generation**
> Yes, we have included a link to our anonymous repository in the main paper, which has been recently updated. However, due to NeurIPS's new regulations, we are not permitted to include it in the author response. Additionally, we have provided a zip file with code and data in the supplementary materials. We kindly ask the reviewer to check again.
>
> **We sincerely appreciate the reviewer’s valuable comments and have carefully addressed all concerns in our response. We hope the revisions address your concerns and kindly request reconsideration of the evaluation scores.**

---

### Official Review · Reviewer_yftr · 2025-07-03

**Clarity:** 4
**Significance:** 3
**Originality:** 3
**Rating:** 5
**Confidence:** 4

**Summary:**

The authors propose a method for data augmentation to generate more training data for L2O methods. By solving a relatively small number of problem instances, the authors then use those solutions to generate additional problem instances whose optimal solutions can be efficiently computed using matrix operations on the solution to the original problems. This is accomplished by constructing affine transformations of compact representation of the KKT conditions for the original problem. The authors describe how to efficiently compute these transformations, as well as listing many example transformations that fit the required form. Results are compared against a baseline data augmentation method.

**Questions:**

How does it compare to training on a more diverse data set? It seems as though this method would bias the network towards performing well on problems that are similar to those that have been augmented in the training data. Are there tradeoffs involved, and if so, what are they?

The focus was not on complete coverage of the space but wouldn’t there be an advantage to focusing on that as a goal? That is, what is the motivation for scaling variable coefficients, scaling constraints, etc.?

**Ethical Concerns:**

["NO or VERY MINOR ethics concerns only"]

**Final Justification:**

I think the authors provided reasonable responses to my questions and those of the other reviewers. Therefore, I elect to keep my rating of Accept.

**Limitations:**

Yes

**Paper Formatting Concerns:**

There are a few lines where it appears the font size or kerning may have been adjusted. See, for example, lines 257, 290, 291.

**Quality:**

3

**Strengths And Weaknesses:**

Strengths:
- This paper uses a clever idea to augment the data for L2O methods. Generating training data for these problems is very expensive, and the method is clear, principled, and theoretically grounded.
- The statement of contributions is clear, and represents a small, but worthwhile step forward in L2O
- The paper is well-written overall, and easy to follow.

Weaknesses:
- The related work section is comprehensive but not particularly useful. It fails to contextualize this work with respect to the citations in that section. The authors need to explain how the work is related to their method, not just that it is related.

Minor:
- Line 93, the edges should be $\subseteq V(G)\times V(G)$
- Line 143 uses $B_{11}$ and $B_{21}$ before they are introduced

---

> ### Author Rebuttal · Authors · 2025-07-30
>
> We sincerely thank the reviewer for the thoughtful and encouraging review. We address your questions and suggestions below.
> ### **Regarding related work**
> Thank you for pointing this out. We agree that our related work section could be improved by better connecting prior work to our method, rather than listing related topics in isolation. In the revised version, we will revise this section to highlight the connections to our work.
> - __MPNNs and Learning to Optimize (L2O)__: In Section 1.1 of the main paper, we review relevant work on MPNNs in the context of L2O. This is directly connected to our setting, as we employ an MPNN architecture on the variable-constraint graph of linearly-constrained quadratic programs (LCQPs). Since LCQPs encompass a wide range of real-world applications (e.g., portfolio optimization, SVMs, Lasso regression), understanding how MPNNs can represent optimization problems and mimic algorithmic behavior is central to our formulation. We also discuss theoretical aspects to ground this connection.
> - __Graph data augmentation__: We include a review of classical graph data augmentation techniques, such as edge perturbation, node dropping, and subgraph sampling. However, we point out their limitations in our specific setting: such augmentations often alter the semantic meaning of optimization problems or violate feasibility. This distinction is also verified empirically, as shown in our experiments, where general-purpose augmentations underperform compared to our mathematically principled ones.
> - __Graph representation learning__: In Appendix C, we provide a background on graph contrastive learning and related representation learning frameworks. This is important because our augmentations are ultimately designed to be plugged into a contrastive pretraining pipeline. We compare these approaches in Table 2 and demonstrate that contrastive learning is only effective when paired with augmentations that respect the structure of the optimization problem. Besides, generic graph representation learning methods often fail to capture meaningful inductive biases in the L2O setting.
> - __Instance generation for MILPs__: We also review recent work on generating in-distribution instances for MILPs. While at first glance similar, these methods differ from ours in several key aspects. Their goal is to learn generative models that fit the instance distribution, often disregarding feasibility or boundedness. Some methods rely on strong structural assumptions, e.g., diagonalizable constraint matrices. Besides, solving the generated problems still requires running a solver, incurring additional computational cost. In contrast, our method is solver-free, learning-free, and mathematically grounded. By leveraging known solutions and applying solution-preserving transformations to the KKT conditions, we can recover solutions to new instances analytically. This makes our approach not only computationally efficient but also interpretable and principled. Moreover, our method is tailored to convex problems (specifically LCQPs), while MILP instance generation targets fundamentally different, combinatorial optimization problems.
>
> ### **Training on a diverse dataset**
> Thank you for the insightful suggestion.
> __Self-supervised setting__: To explore the effect of pretraining on more diverse datasets, we extended the experiments in Table 2 by introducing additional pretraining setups. These new instances vary in both **instance size** and **problem distribution**. Note that generating data is not expensive, because the pretraining is self-supervised and doesn't require solving for the primal-dual solutions. This means we can generate large datasets without solving any optimization problems. Specifically:
> -   We pretrained one MPNN on large-scale random QP instances with 1000 to 1500 variables and constraints—approximately 100–200 times larger than those used in Table 2.
> -   We pretrained another model on a mixed dataset of similar size, containing randomly generated QPs, soft-margin SVMs, and Lasso regression problems.
>
> We observed that contrastive pretraining on diverse datasets __converges significantly faster__, indicating that the contrastive task is easier. This is expected, as in this case, negative samples in contrastive loss are from different classes and are trivially distinguishable, encouraging the model to learn **superficial shortcuts** rather than meaningful representations. Similar phenomena have been observed in [1] as too easy constrastive task is not beneficial.
> After pretraining, we fine-tuned all models on the original random QP dataset used in Table 2. The results are as follows:
>
> | Pretrained | 10% data | 20% data | 100% data |
> |--------|-----|-----|----|
> | No | 5.304±0.229 | 3.567±0.034 | 1.240±0.088 |
> | On QP100 | 3.791±0.097 | 2.427±0.083 | 0.926±0.031 |
> | On QP1000 | 3.632±0.139 | 2.750±0.079 | 1.325±0.159|
> | On mixed |__8.628±0.618__|__4.821±0.503__|__1.419±0.075__|
>
> The number in bold font are the __worst__ performing. Interestingly, pretraining on a highly diverse dataset results in worse performance than no pretraining at all. This is because the model learns from too easy contrastive task in data heterogeneous setting, and fails to capture useful data representations. This leads to poor transferability.
>
> __Supervised setting__: To evaluate the effect of augmentation supervised learning, we construct a mixed dataset of similar scale to Table 1 (approximately 100 variables and constraints), comprising 10000 diverse QP instances including randomly generated QPs, soft-margin SVMs, and Lasso regression problems. Due to limited time for author response, we employ inactive constraint dropping as a representative augmentation method, without hyper parameter tuning. The results are summarized below:
>
> | Augmentation| 10% data | 100% data |
> |--------|-----|----|
> | No | 12.982±0.701 |  4.372±0.310 |
> | Drop cons. | __11.778±0.519__ |  __3.896±0.275__ |
>
> This preliminary result indicates that augmentation has a positive effect in the supervised setting as well. We plan to conduct a more comprehensive evaluation for the camera-ready version.
>
> ### **Regarding trade-off of our method**
> We acknowledge the trade-off introduced by our augmentation approach. One immediate drawback is the additional computation required to generate augmented instances. However, as discussed in Appendix F.1, this overhead can be effectively mitigated by asynchronously prefetching the next batch during model training, resulting in minimal impact on overall training time.  Another trade-off lies in training convergence: when performing supervised learning with augmentations—especially under high perturbation rates—convergence can be slower. This is expected, as the model must generalize over a broader distribution of instances. Nevertheless, we believe this added diversity ultimately leads to more robust generalization.
>
> ### **Not covering the whole design space**
> Thank you for raising this point. Equation (7) in the main paper presents a **universal formulation** that, in principle, can represent a broad class of transformations over the solution space. However, as discussed in the subsequent section, this general form comes with meaningful **practical constraints**. To navigate these trade-offs, we introduce two key concepts: _efficiently recoverable transformation_, transformations for which solutions can be computed efficiently, relying solely on analytic operations; and _solution-independent transformation_, those that preserve feasibility and optimality without requiring access to the original solution. These criteria reflect our design philosophy: to create augmentations that are not only effective, but also computationally efficient, theoretically grounded, and easy to implement in practice.
>
> ### **Typos**
> Thank you for pointing out the typos and formatting issues. We have corrected them and will ensure that the camera-ready version is clean and refined.
>
> **We sincerely thank the reviewer for the constructive feedback and insightful questions. We hope our responses have addressed the concerns raised and clarified the strengths and contributions of our work.**
>
> [1] You, Yuning, et al. "Graph contrastive learning with augmentations." Advances in neural information processing systems 33 (2020): 5812-5823.

---

### Official Review · Reviewer_55gp · 2025-07-03

**Clarity:** 2
**Significance:** 3
**Originality:** 3
**Rating:** 5
**Confidence:** 3

**Summary:**

This paper introduces a principled approach to data augmentation tailored for QPs via MPNNs. The method leverages theoretically justified data augmentation techniques to generate diverse yet optimality-preserving instances. The paper presents various forms of data augmentation for QP problems and provides proofs to ensure the validity of the generated data. The value of the proposed algorithm is demonstrated from multiple perspectives, including improvements in supervised learning, supervised finetuning, generalization, and practical computational overhead.

**Questions:**

Have you considered extending the method to non-convex QPs or other non-convex optimization problems? How does the convexity assumption limit practical applications currently?

Are there plans to test the method’s performance on more real-world datasets?

is there a way to achieve fully solution-independent data augmentation?

Why was the temperature parameter $\tau = 0.1$ chosen for contrastive learning? Have you conducted sensitivity analysis on this parameter?

**Ethical Concerns:**

["NO or VERY MINOR ethics concerns only"]

**Final Justification:**

I have read the authors' rebuttal and believe that the issues that could be addressed have been satisfactorily answered. There are also some issues (e.g., the non-convex case) that I certainly do not expect to be resolved in this paper, which is acceptable. After reviewing the comments from other reviewers and their exchanges with the authors, I still consider this a strong paper and will maintain my original evaluation.

**Limitations:**

yes

**Paper Formatting Concerns:**

-

**Quality:**

3

**Strengths And Weaknesses:**

This is a mathematically grounded data augmentation framework for LPs and QPs, and detailed derivations and proofs for their augmentation framework.
The paper provides extensive experiments.
The framework emphasizes computational efficiency, and the paper discusses how structured matrices (e.g., diagonal matrices) reduce computational overhead, which is supported by runtime experiments.

The experiments focus on synthetic datasets for LPs and QPs, which may not generalize to real-world optimization problems.
While the authors acknowledge this limitation, the lack of results on benchmarks reduces the practical impact of the work.
Some proposed transformations require access to the primal-dual optimal solutions of the original problem, which may not always be available in practice. This could limit the framework’s applicability.
The mathematical sections (e.g., Section 2.1) are dense and may be difficult for readers without a strong optimization background to follow.
Only applied to convex problens.
Experiments focus on synthetic datasets.
The experimental results lack sufficient discussion in the main text.

---

> ### Author Rebuttal · Authors · 2025-07-30
>
> We sincerely thank the reviewer for the thoughtful and constructive feedback, which has helped us improve the clarity and depth of our work. Below we provide point-by-point responses to your concerns.
> ### **Regarding real-world dataset**
> We fully acknowledge this limitation. We do an evaluation on QPLIB. Due to the limited size, large scale, and extreme heterogeneity of QPLIB instances, performing a standard train/validation/test split is infeasible. Existing works that use QPLIB for evaluation typically rely on perturbing problem coefficients [1][2]. These approaches, however, have significant limitations:
> 1. The augmentations are not label-preserving and thus require solving each perturbed instance from scratch;
> 2. Perturbations may break feasibility, which limits the perturbation strength;
> 3. As a result, weaker perturbations are often used, but they yield training instances that are nearly identical to the test instances, making evaluation less meaningful.
>
> Our proposed method, on the other hand, includes both structural and featural perturbation and is targeted at better generalization performance. For the current perturbation method and ours to work well, one can reduce the perturbation rate to an arbitrarily small value to fit perfectly on the test data, but that is meaningless. To study transferability and fitting efficiency on QPLIB, we conduct a pretraining experiment:
> -  Step 1: We generate a foundation dataset of 10000 large-scale random QP instances with sizes ranging from 1000 to 1500 variables and constraints, being 100–200 times larger than the problems used in Table 2. Since no labels are needed for contrastive pretraining, data generation is efficient. Training takes around 30 seconds per epoch using 4 NVIDIA L40S GPUs.
> -  Step 2: We select feasible LCQP instances from QPLIB that vary a lot in size, relax integer constraints, and train an MPNN to fit these problems in a supervised setting. We compare models trained from scratch with models initialized from the pretrained weights.
>
> Here are the problem statistics and results:
>
> | Name | cons. | vars. | A density | Q density |
> |-------------|--------|-------|-----------|-----------|
> | QPLIB_3694 | 3280 | 3240 | 0.001208 | 0.000313 |
> | QPLIB_3708 | 12917 | 12930 | 0.000171 | 0.000628 |
> | QPLIB_3861 | 4650 | 4530 | 0.000856 | 0.000222 |
> | QPLIB_3871 | 1040 | 1025 | 0.003772 | 0.001000 |
> | QPLIB_8559 | 5000 | 10000 | 0.000500 | 0.000700 |
>
> We train each model for 100 epochs and compare predicted objectives to the ground-truth optimal value. Longer training will diminish the advantage of pretraining, as we are testing exactly on the training data. We repeat the pretraining and fine-tuning with three random seeds and report the mean.
>
> | ID | Pretrain | Obj.(optimal) | Obj.(predict) |
> |-------------|----------|---------------|---------------|
> | QPLIB_3694 | No | 0.0 | 10.36161 |
> | QPLIB_3694 | Yes | 0.0 | __0.14176__ |
> | QPLIB_3708 | No | -42.46981 | __-42.46981__ |
> | QPLIB_3708 | Yes | -42.46981 | -42.46916 |
> | QPLIB_3861 | No | 0.0 | 10.89544 |
> | QPLIB_3861 | Yes | 0.0 | __0.41946__ |
> | QPLIB_3871 | No | 0.0 | 7.20569 |
> | QPLIB_3871 | Yes | 0.0 | __-0.19335__ |
> | QPLIB_8559 | No | 15.79288 | 29.20809 |
> | QPLIB_8559 | Yes | 15.79288 | __12.33148__ |
>
> As shown, pretrained models generally fit faster and more accurately within limited training epochs. QPLIB_3708 is an exception, though both models converge very close to the actual objective. These results support the scalability and transferability of our method.
> [1] Wu, Chenyang, et al. "On representing convex quadratically constrained quadratic programs via graph neural networks." arXiv preprint arXiv:2411.13805 (2024).
> [2] Yang, Linxin, et al. "An Efficient Unsupervised Framework for Convex Quadratic Programs via Deep Unrolling." arXiv preprint arXiv:2412.01051 (2024).
>
> ### **Regarding label requirement in augmentations**
> We agree with the reviewer that certain augmentations do rely on access to both primal and dual solutions. Indeed, these augmentations are useful in supervised settings. However, for self-supervised pretraining, we also propose several label-free augmentations, such as scaling the coefficients of variables and scaling constraints, adding redundant constraints, and dummy variables. These augmentations are designed to be solution-independent and are shown to perform well even when applied directly in the supervised setting (see Table 1 in the main paper).
> That said, we would like to highlight that achieving fully solution-independent augmentation is intrinsically difficult. First of all, the most universal form is Equation (7) in the paper. If we consider applying the bias terms, they must satisfy $$\mathbf{B} \left(\mathbf{M}^T\right)^{\dagger} \begin{bmatrix} \mathbf{x}^* \\\ \boldsymbol{\lambda}^* \end{bmatrix} = \mathbf{\beta}$$ which requires the primal and dual solutions. Second, the $\mathbf{M}_{22}$ matrix has its restriction as we wrote in Proposition 2.1, so that we should be careful not to violate the inequalities, which also requires the slack variables solution. If those requirements are met, e.g., the constraints are all equalities, given the form $$\mathbf{M} \begin{bmatrix} \mathbf{Q}  &\mathbf{A}^T  \\\ \mathbf{A}  &  \mathbf{0} \end{bmatrix} \mathbf{M}^{\dagger} \begin{bmatrix} \mathbf{x}^* \\\ \boldsymbol{\lambda}^* \end{bmatrix} = \mathbf{M} \begin{bmatrix} -\mathbf{c} \\\ \mathbf{b} \end{bmatrix}$$
> we can retrieve the new solution with $\mathbf{M}^{\dagger}$. However, _fully solution-independent_ means it is not to compute the new solution, but instead to be applied in self-supervised learning. In the contrastive learning setting, label-free augmentations must retain the same semantic meaning. Arbitrary transformations using general $\mathbf{M}$ may distort the instance such that it no longer corresponds to the same underlying solution, undermining the contrastive objective. To this end, we introduce and formally define the notions of _efficiently recoverable transformation_ and _solution-independent transformation_, which help guide augmentation design in a principled, effective, and pragmatic way.
> ### **Application on non-convex problems**
> We agree with the reviewer that our current framework is restricted to linearly-constrained quadratic programs (LCQPs). While this is a limitation, we note that LCQPs encompass a broad range of real-world problems, such as soft-margin SVM, Lasso regression, and portfolio optimization, which we include in our experiments.
> Our theoretical foundation relies on KKT conditions, which are both necessary and sufficient for convex problems. In non-convex settings, however, the KKT conditions are only necessary (i.e., not sufficient) for local optima and do not apply to global minima. Thus, extending our augmentation theory and instantiations to non-convex optimization is highly non-trivial. Nonetheless, we agree this is an important direction for future work.
> ### **Dense mathematics in Section 2.1 and limited discussion of experimental results**
> Thank you for pointing this out. We acknowledge that the mathematical exposition in Section 2.1 may be dense and difficult to follow due to space constraints. In the camera-ready version, we will use the additional page to enhance the narrative and provide a clearer and more intuitive explanation of the theory and its connection to the augmentations and experiments.
> ### **Temperature tuning in the contrastive loss**
> Thank you for the sharp observation. We conducted ablation experiments over the temperature $\tau$ in the contrastive loss on a random LP dataset. We use the same experiment setting as in Table 2, and run our method with different $\tau$ in {0.01, 0.1, 1.0}. As can be seen in the table below, 0.1 is consistently the best choice, as shown in our Table 2 in the paper. Since our paper is not focused on contrastive learning per se, we did not explore more advanced techniques such as temperature scheduling, but we agree they could be interesting avenues for future work.
>
> | tau | LP 10% | LP 20% | LP 100% |
> |------|-------------|---------------|------------|
> | 0.01 | 3.483±0.074 | 2.903±0.078 | 1.671±0.053 |
> | 0.1 | __3.472±0.086__ | __2.794±0.049__ | __1.588±0.056__ |
> | 1. | 4.269±0.197 | 2.802±0.032 | 1.628±0.046 |
>
> **Once again, we sincerely appreciate the reviewer’s insightful comments and hope our responses address your concerns.**

---

### Note · Authors · 2025-08-11

We would like to sincerely thank the reviewers for their dedicated efforts and the high quality reviews. We are very pleased that the reviewers unanimously recognized the novelty and significance of our proposed idea, along with the paper's solid technical contributions.

During the rebuttal period, we carefully considered every comment and concern raised. The reviewers raised several valuable points for discussion, which primarily focused on more experiment settings, and wider applicability to optimization problems. We have addressed all of these points, specifically:
- We added new experiments to further validate our claims, including more hyperparameter tuning, different augmentation intensities, training on diverse datasets, supervised pretraining, and application to real-world dataset (QPLIB).
- We have provided clearer explanations of our wide application domains, and their relationship to the underlying mathematical principles.
- We have elaborated our related work section to more clearly delineate our contributions. Particularly, in contrast to MILP in-distribution generation papers, we have highlighted both the novelty of our problem formulation, and the unique theoretical and methodological contributions.

We made every effort to directly address each concern in detail. We believe we have sufficiently addressed the reviewers' remarks and suggestions.

We thank the reviewers once again for their engagement and valuable feedback.

---

### Decision · Program_Chairs · 2025-09-17

**Decision:**

Accept (spotlight)

**Comment:**

The paper presents a data augmentation approach for quadratic programming problems for use in machine learning approaches for optimization. Reviewers unanimously agree on the merits of the work, find it clever, with strong experimental results, nice mathematical foundation. The extensive rebuttal and discussion phase is noted, that successfully addressed raised concerns w.r.t. generalizability.
The paper meets the bar for acceptance at NeurIPS and might be of broader interest for the ML4OPT community.